# C-JUN overexpressing CAR-T cells in acute myeloid leukemia: preclinical characterization and phase I trial

Shiyu Zuo[1,2,13], Chuo Li[1,2,3,13], Xiaolei Sun[1,2,13], Biping Deng[4], Yibing Zhang[1,2], Yajing Han[1,2], Zhuojun Ling[5], Jinlong Xu[5], Jiajia Duan[5], Zelin Wang[5], Xinjian Yu[6], Qinlong Zheng[6], Xiuwen Xu[6], Jiao Zong[6], Zhenglong Tian[7], Lingling Shan[1,2], Kaiting Tang[1,2], Huifang Huang[3], Yanzhi Song[8], Qing Niu [1,2], Dongming Zhou [9], Sizhou Feng [1,2], Zhongchao Han[10], Guoling Wang[11] ✉, Tong Wu[8] ✉, Jing Pan [12] ✉ & Xiaoming Feng [1,2,3] ✉

Chimeric antigen receptor (CAR) T cells show suboptimal efficacy in acute myeloid leukemia (AML). We find that CAR T cells exposed to myeloid leukemia show impaired activation and cytolytic function, accompanied by impaired antigen receptor downstream calcium, ZAP70, ERK, and C-JUN signaling, compared to those exposed to B-cell leukemia. These defects are caused in part by the high expression of CD155 by AML. Overexpressing C-JUN, but not other antigen receptor downstream components, maximally restores anti-tumor function. C-JUN overexpression increases costimulatory molecules and cytokines through reinvigoration of ERK or transcriptional activation, independent of anti-exhaustion. We conduct an open-label, non-randomized, single-arm, phase I trial of C-JUN-overexpressing CAR-T in AML (NCT04835519) with safety and efficacy as primary and secondary endpoints, respectively. Of the four patients treated, one has grade 4 (dose-limiting toxicity) and three have grade 1–2 cytokine release syndrome. Two patients have no detectable bone marrow blasts and one patient has blast reduction after treatment. Thus, overexpressing C-JUN endows CAR-T efficacy in AML.

Acute myeloid leukemia (AML) is a common hematologic malignancy[1]. Despite advances in treatment, there are limited options for salvage therapy in relapsed or refractory (r/r) AML, so the long-term prognosis for r/r AML remains poor. While stem cell transplantation (SCT) may be considered for certain patients[2], outcomes for those who do not achieve complete remission (CR) prior to SCT are dismal[3]. Therefore, there is a significant clinical need for the development of novel targeted therapies to address the challenges of AML.

Chimeric antigen receptor (CAR) T-cell therapy has demonstrated dramatic efficacy in patients with B-cell malignancies, with reported CR rates ranging from 60% to 90%[4]. Despite preclinical studies

suggesting CD33, CLL1, CD123, and other cell surface molecules as target antigens, the feasibility of CAR T-cell therapy in AML has not been fully established[5]. CD33, which is highly expressed on leukemic cells and leukemic stem cells in patients with AML[6], represents an attractive immunotherapy target with a low risk of antigen escape. Nevertheless, studies of CD33 CAR T-cell therapy in r/r AML have reported limited or no significant anti-leukemic effects, even in the presence of cytokine release syndrome (CRS)[7,8]. Similarly, CD123 CAR T-cell therapy showed weak proliferation and low CR rates[9,10], whereas CLL1 CAR T-cell therapy induced anti-leukemic responses in only a small proportion of patients without sustained CAR T-cell

A full list of affiliations appears at the end of the paper. ✉e-mail: wanggl2025@163.com; wut@gobroadhealthcare.com; panj@gobroadhealthcare.com; fengxiaoming@ihcams.ac.cn

expansion[11,12]. Thus, the unsatisfactory efficacy of CAR T-cell therapy in AML appears to be a general phenomenon rather than specific to a particular antigen. However, the exact molecular mechanisms underlying this resistance remain to be elucidated, and strategies to overcome these barriers remain to be identified.

In this work, we aim to elucidate the molecular mechanisms underlying the resistance of myeloid leukemia cells to CAR T-cell therapy and to develop strategies to overcome this resistance. The results suggest that CAR T cells exposed to myeloid leukemia exhibit profound defects in antigen downstream signaling, leading to impaired anti-tumor function. This study also identifies CD155, which is highly expressed on AML, as mediating the suppression of CAR T cell function. Further animal studies confirm that overexpression of antigen receptor signaling elements, specifically C-JUN, can restore the anti-leukemia functionality of CAR T cells. Based on these findings, we conduct a first-in-human Phase I clinical trial to evaluate the safety and efficacy of functionally optimized CD33 CAR T cells in patients with r/r AML. This report presents both in vitro and animal findings as well as preliminary results from the initial four patients in the clinical trial.

## Results

### CAR T cells are less effective at killing myeloid leukemia than B-lineage leukemia

We first compared the anti-tumor activity of CD33 CAR T cells against AML and B-ALL cell lines that were engineered to express equivalent levels of CD33 (U937^CD33 and Nalm6^CD33) (Fig. 1A). CD33 CAR T cells exhibited lower cytotoxicity to U937^CD33 than to Nalm6^CD33 cells at various effector-to-target (E:T) ratios (Fig. 1B). Similar results were observed with CLL1 and CD123 CAR T cells (Fig. 1C). However, there was no significant difference in T-cell counts and the apoptotic rate after co-incubation with U937^CD33 and Nalm6^CD33 (Supplementary Fig. 1A, B). U937^CD33-exposed CAR T expressed fewer activation markers CD25 and CD69, and effector molecules interferon-γ (IFN-γ), granzyme B, and interleukin-2 (IL-2) (Fig. 1D). We further evaluated the cytotoxicity to different leukemia cell lines. The results confirmed a lower killing potency of CD33 CAR T to myeloid U937^CD33 and THP-1^CD33 than to lymphoid Nalm6^CD33 and Raji^CD33 cells. Interestingly, no significant difference in cytotoxicity was observed between Raji^CD33 and HL60^CD33 cells (Fig. 1E). The cytotoxicity of CD38 CAR T to primary AML cells was lower than to primary ALL cells with equivalent levels of CD38 antigen (Fig. 1F and Supplementary Fig. 1C).

Next, we evaluated the anti-tumor activity of CD33 CAR T cells in NSG mice that were pre-inoculated with either U937^CD33 or Nalm6^CD33 cells (Fig. 1G and Supplementary Fig. 1D). Although U937^CD33 and Nalm6^CD33 cells grew at different rates in mice, we observed a faster disease progression and shorter survival in mice bearing U937^CD33 cells at different initial CAR T-to-tumor cell ratios (Supplementary Fig. 1E). CAR T cells were barely detectable in U937^CD33 mice, but were detectable in Nalm6^CD33 mice 2 weeks after CAR T infusion (Supplementary Fig. 1F). These results suggest a severely impaired activity of CAR T cells in mice with myeloid leukemia.

We further evaluated the impact of myeloid cells on CAR T-cell phenotype in vivo by creating conditions where mice had an identical tumor burden before CAR T-cell infusion (Supplementary Fig. 1G, H). After 3–5 days, CAR T cells in mice bearing U937^CD33 tumors had fewer cell counts, higher apoptotic rates, and lower expression of CD25, granzyme B, IFN-γ, and IL-2 than in mice bearing Nalm6^CD33 tumors (Fig. 1H–J and Supplementary Fig. 1I). During an extended observation period of nearly 2 weeks before the disappearance of CAR T cells in U937^CD33 mice, we did not observe any differences in the effector/ memory subsets or exhaustion markers between the two groups (Fig. 1K, L).

### CAR-T has an impaired effector program and defective antigen receptor signal transduction in AML

To investigate the mechanisms underlying CAR T cell incompetence to AML, we co-incubated CAR T cells with U937^CD33 and Nalm6^CD33 cells for 2 days and then sorted CD8^+ CAR^+ T cells to perform RNA-seq (Supplementary Fig. 2A). Gene set enrichment analysis (GSEA) revealed that U937^CD33 co-incubated CAR T cells had a lower level of T cell activation, cytokine production, and T cell exhaustion than Nalm6^CD33 co-incubated cells (Fig. 2A). Interestingly, these cells exhibited reduced T cell receptor (TCR) calcium pathway and extracellular signal-regulated kinase (ERK) 1/2 cascade (Fig. 2A). However, JNK-MAPK and nuclear factor of activated T cells (NFAT) signaling pathways had no significant differences between the two groups (Supplementary Fig. 2B).

We then examined the specific genes differentially expressed in GSEA. U937^CD33 co-incubated CAR T cells had lower expression of cytokines *IL3, IL4, IL22*, and *IL23A*, cytotoxic molecules *IFNg, GNLY, GZMB*, and *GZMH*, and costimulatory molecules *TNFRSF4 (OX40), TNFRSF9 (4-1BB), CD80, TNFRSF18 (GITR)*, and had lower exhaustion-related genes including *PDCD1 and LAG3* (Fig. 2B). Despite a lower level of IL-2 protein in AML-co-incubated CAR T cells, *IL2* mRNA showed no significant change, suggesting that IL-2 may be subject to post-transcriptional regulation. U937^CD33 co-incubated CAR T cells also showed decreased expression of calcium signaling-related *CSF2*[13], and *IFNg*[14], and the positive regulators of ERK signaling including *PDGFA*[15], *NTRK1*[16], *MTURN*[17], *IGF1*[18], and *CFLAR*[19], and increased expression of *DUSP1*[20], a critical phosphatase that inactivates ERK (Fig. 2B).

We next assessed the activity of antigen receptor downstream signals. Upon antigen ligation to TCR, proximal lymphocyte-specific protein tyrosine kinase (LCK) and zeta-chain-associated protein kinase 70 (ZAP70) signaling are triggered, leading to calcium mobilization, ERK, and JNK activation, and subsequent activation of transcription factors such as NFAT, FOS, and C-JUN[21,22]. However, the phosphorylation of ERK in CAR T cells co-incubated with U937^CD33 cells for 15–30 min is not different from that of Nalm6^CD33 cells, suggesting no immediate defect in CAR T signaling after a short AML exposure (Supplementary Fig. 2C). To gain further insight into signal transduction, we pre-incubated CAR T cells with U937^CD33 and Nalm6^CD33 cells for 12 h and then sorted these cells and re-stimulated them with U937 cells. This two-step protocol is indeed to study the reprogramming events that occur within CAR T cells during the initial 1-h co-incubated with myeloid and lymphoid tumor cells. We found that U937^CD33 pre-incubated CAR T cells exhibited less calcium influx and lower phosphorylated ZAP70 (p-ZAP70), ERK1/2 (p-ERK1/2), and C-JUN (p-C-JUN) proteins compared to Nalm6^CD33 pre-incubated CAR T cells (Fig. 2C–F). Conversely, phosphorylated JNK protein did not show a difference (Fig. 2G).

U937^CD33 pre-incubated CAR T cells that received second stimulation with PMA or anti-CD3 had lower p-ERK and p-C-JUN than Nalm6^CD33 pre-incubated CAR T cells (Supplementary Fig. 2D, E). U937^CD33 pre-incubated CAR T cells had lower levels of p-ZAP70 after anti-CD3 stimulation, but p-ZAP70 was not different after PMA stimulation, which bypassed TCR (Supplementary Fig. 2F). Total ERK and C-JUN proteins in U937^CD33 co-incubated CAR T cells are comparable to or higher than those in Nalm6^CD33 co-incubated CAR T cells, while total ZAP70 protein was slightly lower (Supplementary Fig. 2G, H).

### AML-expressed CD155 inhibits CAR T killing and attenuates the ERK signaling pathway

We next explored which AML-derived factors are responsible for the impaired function of CAR T cells. Reactive oxygen species (ROS) and nitric oxide (NO), the known immunosuppressors[23,24], were expressed at varied levels across different AML and ALL cell lines (Supplementary Fig. 3A, C). Primary AML samples showed higher expression of ROS but not NO (Supplementary Fig. 3B, D). However, inhibition of NO and ROS

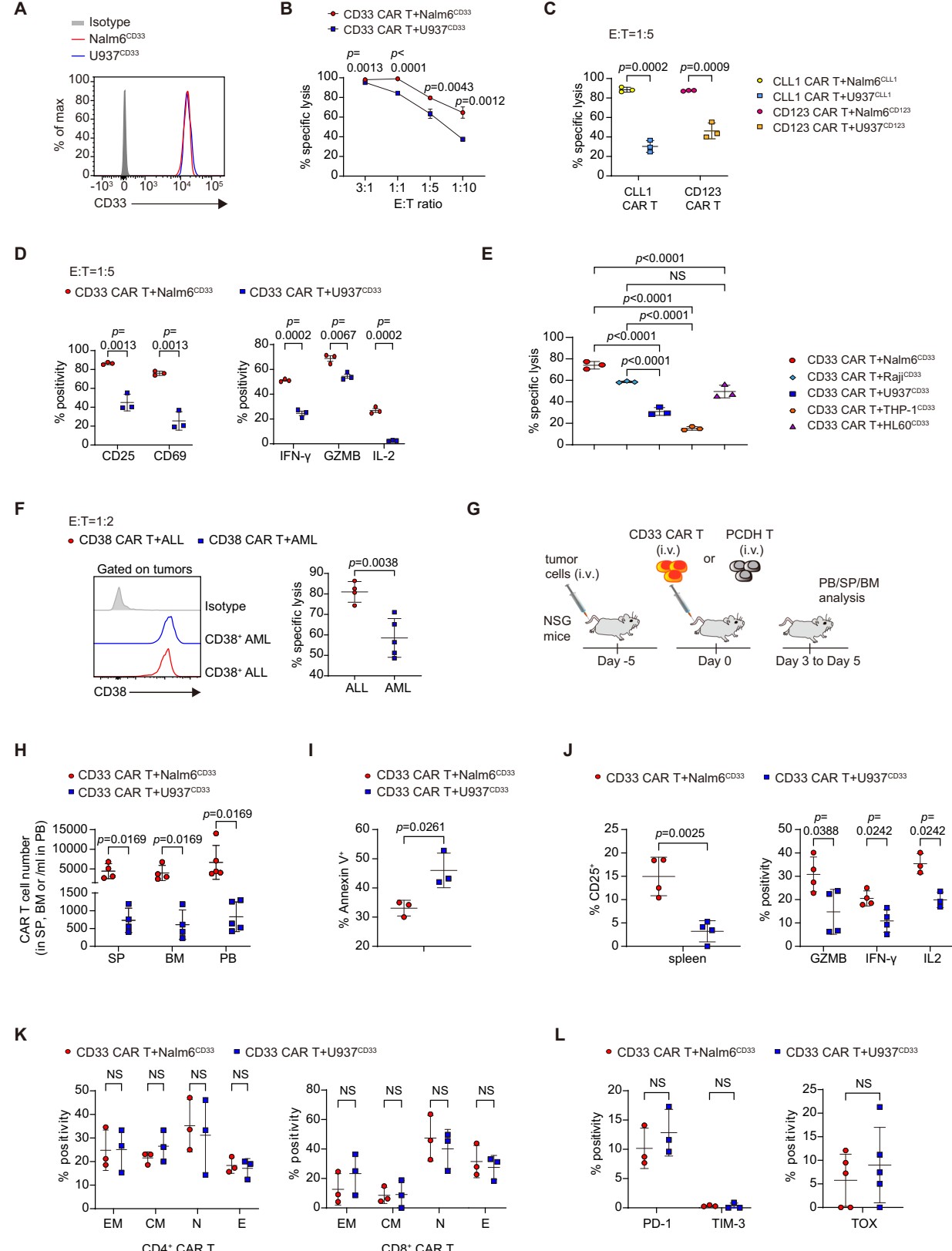

did not improve CAR-T cytotoxicity to U937[CD33] cells (Supplementary Fig. 3E). To gain further insight, we performed transcriptome sequencing on several AML and ALL cell lines and primary samples, which revealed higher expression of inhibitory ligands/receptors, including *B7-H3, TIM3*, and *CD155* in AML cell lines and primary samples (Fig. 3A). Because B7-H3[25] and TIM3[26] were inhibitory receptors, their

expression on leukemia cells were unlikely to deliver an inhibitory signal in CAR T cells. Therefore, we focused on CD155 which serves as an inhibitory ligand.

Consistently, AML cell lines and primary samples had higher levels of CD155 protein than lymphoid controls (Fig. 3B). CD155 blocking antibody enhanced the killing activity of CAR T cells against U937 cells

**Fig. 1 | CAR T cells are less effective at killing myeloid leukemia than B-lineage leukemia. A** Representative histograms showing CD33 expression. **B** Cytolytic activity of CD33 CAR T cells in vitro. $n = 3$ biological replicates with T cells from different donors per point. **C** Cytolytic activity of CLL1 and CD123 CAR T cells in vitro, $n = 3$. **D** Percentage of CD25 and CD69, IFN-γ, Granzyme B (GZMB), and IL-2 in CAR T cells, $n = 3$. **E** Cytolytic activity of CD33 CAR T cells against different cell lines in vitro, $n = 3$. **F** Representative histograms showing CD38 expression on sorted primary samples and CD38 CAR T cytolytic activity in vitro, $n = 4$ in ALL group, $n = 5$ in AML group. **G** Schematic of mouse model. NSG mice were intravenously injected with tumor cells, followed by $1 \times 10^6$ CAR or PCDH T cells 5 days later. Peripheral blood (PB), spleen (SP), and bone marrow (BM) cells were collected from mice euthanized on days 3–5. **H** CAR T-cell counts in SP, BM, and PB from (**G**), $n = 4$ in SP and BM, $n = 5$ in PB. **I** Percentage of Annexin V⁺ of CAR T cells in spleen from (**G**), $n = 3$. **J** Percentage of CD25, GZMB, IFN-γ, and IL-2 of CAR T cells in spleen from (**G**), $n = 3$ in IL-2 expression, $n = 4$ otherwise. **K** Total Naïve (N, CD45RA⁺CD62L⁺), central memory (CM, CD45RA⁻CD62L⁺), effector memory (EM, CD45RA⁻CD62L⁻) and effector (E, CD45RA⁺CD62L⁻) CAR T cells were assigned to CD4⁺ and CD8⁺ CAR T cells in spleen, $n = 3$. **L** Percentage of PD-1, TIM-3, and TOX in CAR T cells in spleen, $n = 5$ in TOX, $n = 3$ otherwise. For all bar plots, data are shown as mean ± SD. Assays were performed on day 10 after T-cell initial activation. Two-sided unpaired $t$-tests or multiple two-sided unpaired $t$ tests were used to assess significance in (**B, C, D, F, H–L**). One-way ANOVA was used in (**E**). All numbers defined by "$n$" indicate the number of biological replicates with different human donors or mice. Data are representative of two independent experiments. NS not significant. Source data are provided in the Source Data file.

(Fig. 3C). When CD155 was knocked out in U937 cells, the killing activity of CAR T cells was partially restored (Fig. 3D, E), and the levels of p-ERK and p-C-JUN increased, while p-ZAP70 showed no change (Fig. 3F and Supplementary Fig. 4A). However, CD155 did not affect the total ERK, C-JUN, or ZAP70 protein expression (Supplementary Fig. 4B). Besides, overexpression of CD155 in Nalm6 cells decreased the cytotoxicity of CD19 CAR T cells (Fig. 3G).

CD155 and the closely related CD112 are ligands for stimulatory receptor CD226 and inhibitory receptors TIGIT and CD96[27,28]. CD112 expression varied across AML and ALL cell lines, although its expression was higher in U937 than in Nalm6 (Fig. 3H). Blocking CD112 did not affect the cytotoxicity of CD33 CAR T to U937 (Fig. 3I). However, antibody blockade or knockout of TIGIT, CD96, or CD226 did not affect the cytotoxicity of CD33 CAR T cells (Fig. 3J, K and Supplementary Fig. 4C, D, F). Furthermore, blocking both TIGIT and CD96 simultaneously had no effect on CD33 CAR T cell-cytotoxicity (Fig. 3L and Supplementary Fig. 4E).

### Overexpression of C-JUN restores CAR T function in AML
We next sought to define a strategy to restore CAR T function in AML. Because the blockade of TIGIT and CD96 was unable to rescue the functionality of CD33 CAR T cells, we sought to restore CAR T function by overexpressing key antigen receptor signaling elements to improve therapeutic efficacy. LCK and ZAP70 provide T cell receptor proximal signaling for antigen downstream signal initiation and transduction[29]. C-JUN and C-Fos are members of the AP-1 complex, which regulates T cell activation and effector function[30]. Thus, we developed CD33 CAR constructs overexpressing LCK, ZAP70, C-JUN, C-Fos, and IFN-γ, and that overexpressing IL-15, a cytokine known to potentiate CAR T therapeutic activity in various tumor models[31–33], as a control (Fig. 4A, B and Supplementary Fig. 5A, B). The activity of CAR T cells overexpressing IL-15, LCK, ZAP70, and C-JUN, but not C-Fos and IFN-γ, improved at various degrees (Fig. 4C–E and Supplementary Fig. 5C, D). CAR T cells overexpressing IL-15 showed enhanced expansion in mice but unfortunately did not control tumor progression as effectively as other molecules (Fig. 4C–F). CAR T cells overexpressing LCK effectively controlled tumor progression at an early stage, but the effect was not as long-lasting as that of ZAP70 or C-JUN, whereas overexpression of C-JUN showed the greatest effect in improving tumor control and prolonging mouse survival (Fig. 4C–E). C-JUN overexpression did not affect the CAR transduction rate (Supplementary Fig. 6A–D), and C-JUN CAR T cells showed superior efficacy despite different transduction rates in different experimental batches (Supplementary Fig. 6E, F). C-JUN effect was recapitulated in CD123 CAR T cells (Fig. 4G–I).

C-JUN overexpression alone in non-CAR T cells could not eliminate tumors or prolong survival (Fig. 4J–L). We next evaluated C-JUN impact on CAR T cells with different costimulation domains and observed that mice in the CD33 bbz-C-JUN and CD33 28z-C-JUN CAR T cell treatment groups had comparable effects in enhancing tumor control compared with non-C-JUN CAR T cells (Fig. 4M–O and

Supplementary Fig. 6H). In C-JUN CAR T-treated mice, despite enhanced anti-tumor activity, CAR T cells gradually decreased over time (Supplementary Fig. 6I), leading to tumor progression and eventual death in a proportion of mice. C-JUN CAR T cells did not induce a significant increase in cytokine levels in tail blood (Fig. 4P), and pathological evaluation of vital organs revealed no significant inflammatory cell infiltration, suggesting that C-JUN CAR T did not cause inflammatory organ toxicity in mice model (Fig. 4Q).

We next investigated the effect of C-JUN overexpression on CAR T-cell expansion and function in vivo. C-JUN CAR T cells sorted from U937-bearing mice exhibited greater cytotoxicity against U937 than control CAR T cells (Fig. 4R). Compared to control groups, C-JUN CAR T cells had higher in vivo numbers and showed a higher proliferative rate (Fig. 5A, B). They expressed higher IL-2 receptor (CD25) and IFN-γ (Fig. 5C, D). However, no significant difference in the expression of IL-2 was observed (Fig. 5D). Contrasting to control cells which were mainly naïve subpopulations, C-JUN CAR T cells were mainly enriched in effector memory subpopulations within the CD4⁺ subset and enriched in both effector memory and central memory subpopulations within the CD8⁺ subset (Fig. 5E and Supplementary Fig. 6J). C-JUN CAR T cells did not show significantly changed apoptosis (Supplementary Fig. 6K) or alleviated exhaustion (Fig. 5F).

C-JUN CAR T cells maintained in vitro also exhibited enhanced killing ability and produced more IL-2 and IFN-γ when co-incubated with U937 (Fig. 5G, H). We next tested whether C-JUN overexpression had any effect on the tonic signal in CAR T cells without antigen stimulation. We found no significant differences between control CAR T cells and C-JUN CAR T cells cultured alone in terms of apoptosis and inhibitory receptors (PD-1, TIM-3, LAG-3) (Fig. 5I, J). There was an increase in CD3ζ phosphorylation (p-CD3ζ) in both control CAR T cells and C-JUN CAR T cells compared to PCDH T cells, but no difference was found between control CAR T cells and C-JUN CAR T cells (Fig. 5K). It should be noted that both CAR and TCR can induce CD3ζ phosphorylation, so these results suggest that C-JUN overexpression had little effect on the overall CD3ζ signaling downstream of both TCR and CAR in the absence of overt antigen stimulation.

### Overexpression of C-JUN upregulates costimulatory molecules and cytokines and reactivates ERK
To understand the mechanisms underlying the enhancing effect of C-JUN, we co-incubated CAR T with U937 cells for 2 days, and then a transcriptome analysis was performed on control or C-JUN CAR T cells (Supplementary Fig. 7A). GSEA revealed that C-JUN CAR T cells exhibited higher levels of T cell activation, positive regulation of immune effector process, and ERK1/2 cascade (Fig. 6A). However, no significant changes in JNK-MAPK, NFAT, TCR calcium, or exhaustion pathways were observed (Supplementary Fig. 7B).

Examination of specific genes in C-JUN CAR T cells revealed higher levels of transcripts for cytokines *IL2, IL17F, IL21*, and *IL22*, cytokine receptor *IL1R1*, cytotoxicity mediator *ULBP3*[34], and costimulatory molecules *TNFRSF9 (4-1BB), CD28*, and *CD86* (Fig. 6B). IFNg mRNA is

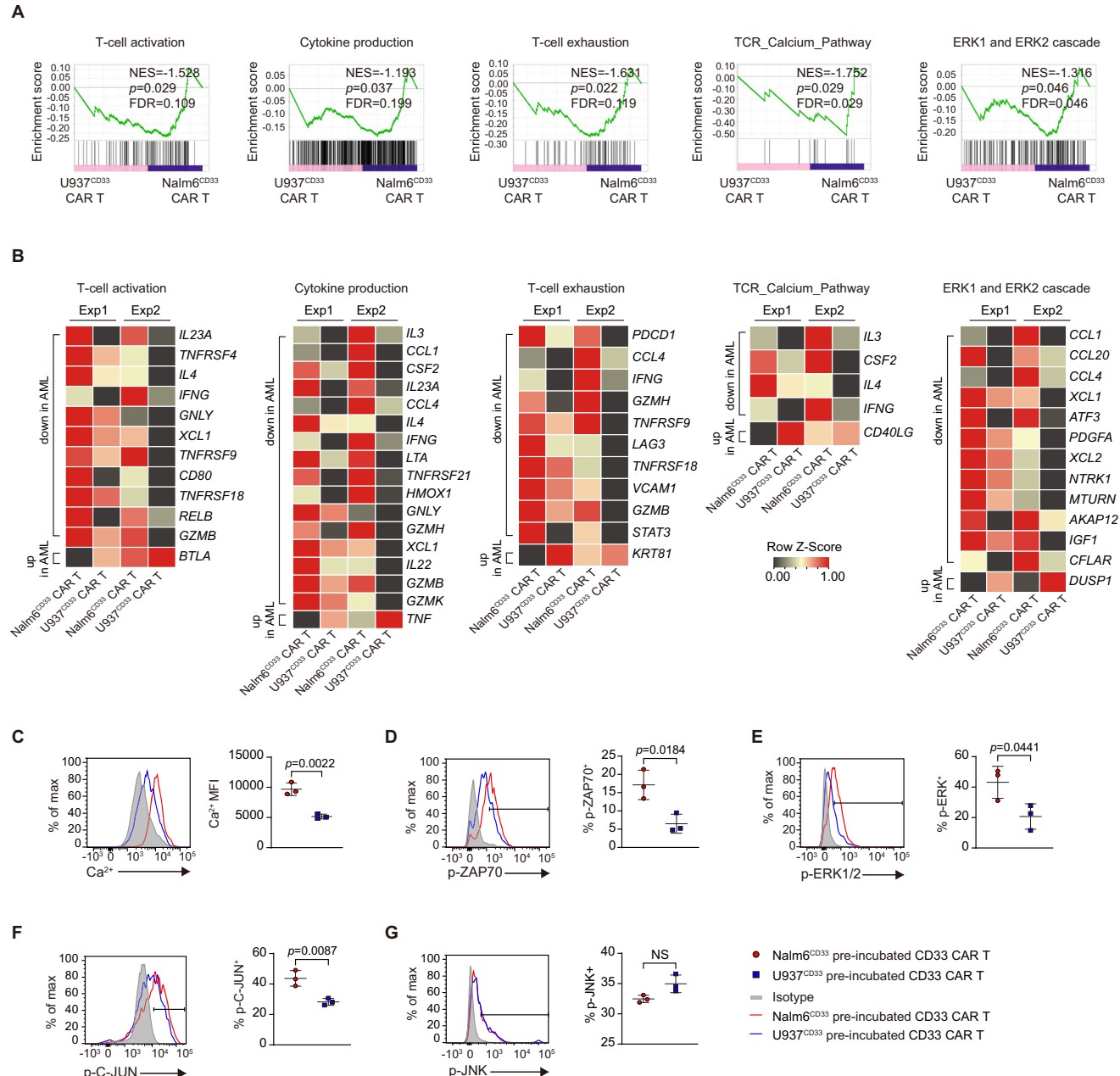

**Fig. 2 | CAR T had an impaired effector program and defective antigen receptor signaling in AML. A**, **B** GSEA results from running RNAseq data of U937$^{CD33}$ co-incubated- versus Nalm6$^{CD33}$ co-incubated-CD33 CAR T cells (**A**). Nominal *P* values, FDR *q* values, and normalized enrichment score (NES) were calculated using GSEA software (Broad Institute). Heat maps (**B**) indicating the expression of genes enriched in GSEA from (**A**) and the known related genes not included in the GSEA gene set. The genes shown in heatmaps meet the parameters: fold change ≥ 1.5-fold in each of the two biological replicates. Each RNA sample was pooled from three technical replicates with T cells from one donor, and we conducted experiment with two different donors, *n* = 2. **C** Intensity of intracellular calcium in CAR T cells

co-incubated with tumor cells, *n* = 3. Percentage of phosphorylated ZAP70 (**D**), phosphorylated ERK1/2 (**E**), phosphorylated C-JUN (**F**), and phosphorylated JNK (**G**) in CD33 CAR T cells pre-incubated with tumor cells and re-stimulated with U937 cells, *n* = 3. For all bar plots, data are shown as mean ± SD. Assays were performed on day 10 after T-cell initial activation. Two-sided unpaired *t*-tests were used to assess significance in (**C**–**G**). All numbers defined by "*n*" indicate the number of biological replicates with different human donors. Data are representative of two independent experiments. NS not significant. Source data are provided in the Source Data file.

also increased in C-JUN CAR T cells, but not more than 1.5 fold (Supplementary Fig. 7C). In addition, C-JUN CAR T cells had higher expression of *CASR*[35]*, EPHB2*[36]*, NRP1*[37]*, PDGFA*[15]*, and PDGFRA*[38], upstream positive regulators of ERK1/2 activation (Fig. 6B). Notably, exhaustion-related genes were mostly upregulated or unchanged in C-JUN CAR T cells (Supplementary Fig. 7C).

In chromatin accessibility analysis, C-JUN CAR T cells had significantly more unique accessible chromatin regions (13192 peaks)

than control cells (7529 peaks). Notably, C-JUN CAR T showed increased chromatin accessibility near genes related to T cell effector function, including *IL1R1* and *CD86*, and the regulators of ERK activation, such as *CASR, NRP1*, and *EPHB2* (Fig. 6C).

Signal transduction activity was then analyzed. After 12 h pre-incubation and 15–30 min re-stimulation with U937 cells, C-JUN CAR T cells had significantly higher levels of p-ERK and p-C-JUN than control CAR T cells, whereas p-JNK showed only a slight increase, and no

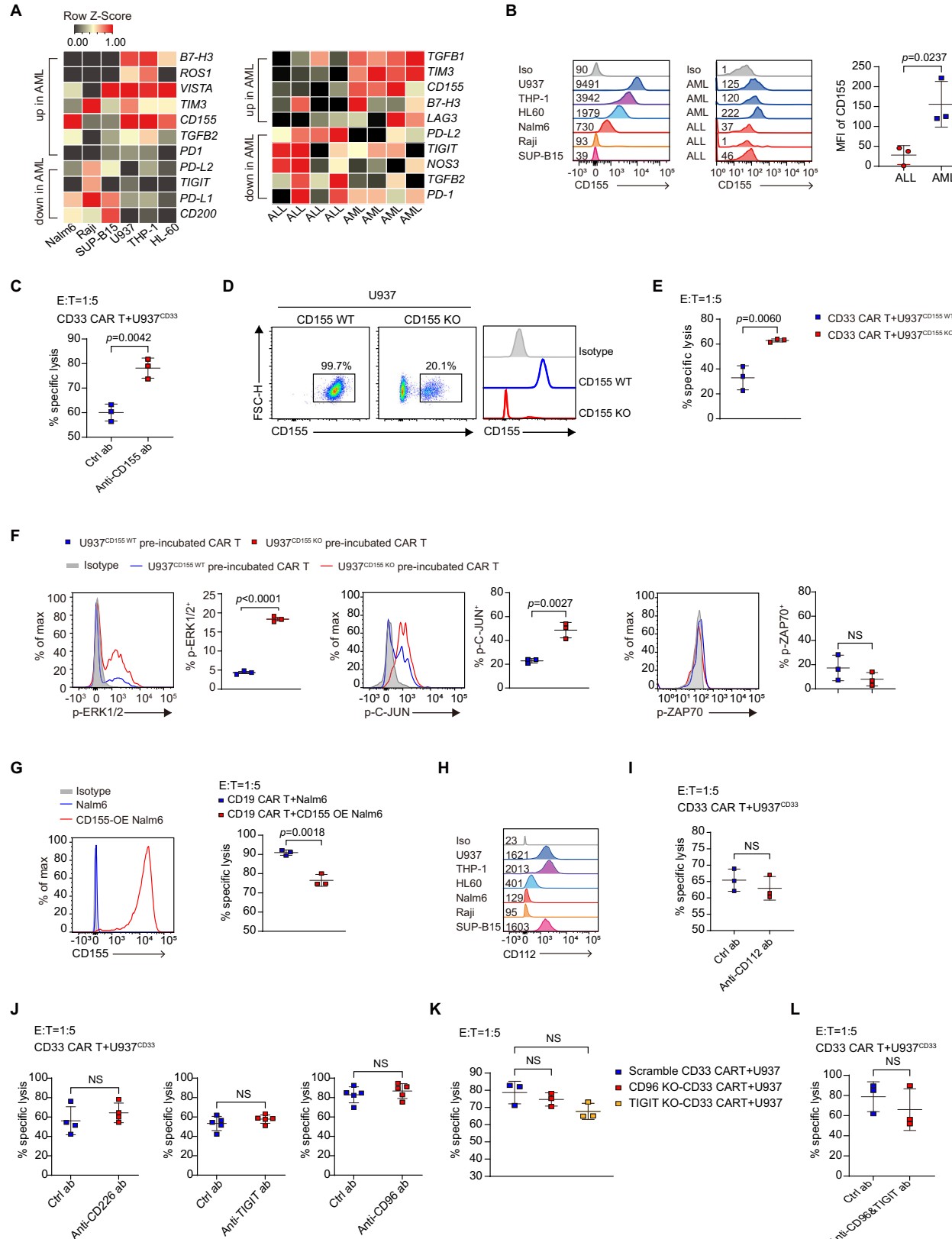

differences in calcium influx or p-ZAP70 were observed (Fig. 6D–H). We also performed a re-stimulation with U937$^{CD33KO}$ (negative control), PMA, or anti-CD3. The results showed that C-JUN CAR T cells re-stimulated with PMA or anti-CD3 or had higher p-ERK1/2, p-C-JUN, and p-JNK (only with anti-CD3) levels than control CAR T cells (Supplementary Fig. 7D). C-JUN overexpression did not change total ERK and

JNK protein levels, but increased C-JUN protein level in CAR T cells that have been co-incubated for 12 h with U937 (Supplementary Fig. 7E). Therefore, the increased p-ERK and p-JNK were unlikely a result of increased total protein levels, but may instead be caused by an increased rate of phosphorylation induced by upstream activating signals.

**Fig. 3 | CD155 mediates myeloid tumor evasion to CAR-T killing. A** Heat maps showing inhibitory genes that are up- or down-regulated more than two-fold. Primary samples each had four biological replicates. **B** Histograms showing CD155 expression of in tumor cell lines and primary samples, and the MFI of CD155 in primary samples, $n = 3$. **C** Cytolytic activity of CD33 CAR T cells against U937 cells in the presence of 500 ng/ml CD155 blocking antibody, $n = 3$. **D** Flow cytometric plots showing knockout efficiency of CD155 on U937 cells. **E** Cytolytic activity of CD33 CAR T cells against U937$^{CD155\ WT}$ and U937$^{CD155\ KO}$ cells, $n = 3$. **F** Percentage of p-ERK1/2, p-C-JUN, and p-ZAP70 in CAR T cells pre-incubated with U937$^{CD155\ WT}$ and U937$^{CD155\ KO}$ cells and re-stimulated with U937 cells, $n = 3$. **G** Expression of CD155 in control and CD155-overexpressing (CD155 OE) Nalm6 cells, and CD19 CAR T cell cytolytic activity against them, $n = 3$. **H** Histograms showing CD112 expression in AML and ALL cell lines. **I** Cytolytic activity of CD33 CAR T cells against U937 cells with 10 μg/ml CD112 antibody, $n = 3$. **J** Cytolytic activity of CD33 CAR T cells against U937 cells with 10 μg/ml CD226, 50 μg/ml TIGIT, or 20 μg/ml CD96 antibody, $n = 4$ in CD226 antibody group, $n = 5$ otherwise. **K** Cytolytic activity of scramble-, CD96 knockout (CD96 KO), and TIGIT knockout (TIGIT KO)-CD33 CAR T cells against U937 cells, $n = 3$. The scrambled CD33 CAR T were electroporated with non-targeting sgRNAs. **L** Cytolytic activity of CD33 CAR T cells against U937 cells with both 50 μg/ml TIGIT and 20 μg/ml CD96 antibody, $n = 3$. For all bar plots, data are shown as mean ± SD. Assays were performed on day 10 after T-cell initial activation. Two-sided unpaired $t$-tests were used to assess significance in (**B, C, E–G, I, J, L**). One-way ANOVA was used in (**K**). All numbers defined by "$n$" indicate the number of biological replicates with different human donors. Data are representative of two independent experiments. NS not significant. Source data are provided in the Source Data file.

After confirming that C-JUN CAR T cells expressed higher levels of 4-1BB, CD28, CD86, and IL-21 protein (Fig. 6I), we investigated whether the increased p-ERK1/2 and p-JNK were involved in the improved cytotoxicity and upregulation of costimulatory molecules/cytokines in C-JUN CAR T cells. The MEK1/2 inhibitor U0126, but not the JNK inhibitor SP600125, abolished the enhanced killing capacity (Fig. 6J). At a high concentration of 50 μM, U0126 suppressed 4-1BB, CD28, IL-2, IL-21, and IFN-γ protein expression, while SP600125 did not affect their protein levels, except for IL-21 (Fig. 6K, L). Furthermore, at a high concentration of 50 μM, U0126 suppressed the expression of 4-1BB, IL-2, IL-21, and IFN-γ in control CD33 CAR T cells. This suggests that ERK is required for the expression of 4-1BB, IL-2, and IFN-γ, whereas JNK and ERK may be involved in the expression of IL-21 (Supplementary Fig. 8A). To further assess whether this elevation of co-stimulatory molecules and cytokines is a result of C-JUN-mediated increase of ERK activity or a direct C-JUN transcriptional regulation, we further titrate down U1026 to a concentration of 5 μM to bring p-ERK levels in C-JUN CAR-T to the same level of control CAR T cells (Fig. 6M). In this condition, U0126 only suppressed 4-1BB, IL-2, and IFN-γ, but not CD28, CD86, and IL-21 protein (Fig. 6N, O and Supplementary Fig. 8B). These results collectively suggest the increased 4-1BB, IL-2, and IFN-γ is likely dependent on C-JUN-mediated increase of ERK activity, but the increased CD28, CD86, and IL-21 protein is rather dependent on the direct C-JUN-mediated transcriptional activation. This suggests that the advantage of C-JUN lies on the synergistic action of feedback reactivation of ERK and its direct transcription factor activity.

### Safety of C-JUN-overexpressing CD33 CAR T cells in patients with r/r AML

We initiated a Phase I trial to evaluate safety and efficacy of optimized CD33 CAR T cells overexpressing C-JUN in patients with r/r AML. Of the 7 patients screened, four were enrolled (Fig. 7A). Two patients were male and two were female. The median age of the four patients enrolled was 9.5 years (range, 3–12). These patients had experienced morphologic relapse without extramedullary disease and had received 2–6 lines of prior therapies, and three had prior transplantation. The baseline and outcomes are in Table 1. All four patients received lymphodepletion before infusion. Four patients received a dose of $0.5 \times 10^6$ (±20%) CAR T cells/kg. These cells were sourced either autologous (the first) or from previous transplantation donors (patients 2, 3, and 4).

Three patients experienced grade 1–2 CRS and one experienced grade 4. The median onset was day 1 post-infusion (range, 1–3), and the median duration was 11 days (range, 6–20) (Fig. 7B, D). One patient also experienced grade 2 CRS on days 2–8 post-second-infusion (Fig. 7B). Two patients received tocilizumab, and all four received corticosteroids and nonsteroidal anti-inflammatory drugs for CRS. The patient with grade 4 CRS presented with fever, dyspnea, and hypotension and did not respond to tocilizumab or corticosteroids, and developed dose-limiting toxicity (DLT), and his serum IL-6, IFN-γ, and TNF-α levels were relatively higher than others (Fig. 7E). Due to elevated TNF-α, etanercept was administered, resulting in relief of symptoms (Fig. 7C). The next two patients also received etanercept to control CRS.

Two patients experienced grade 1 neurotoxicity on day 5 that lasted 1–2 days (Fig. 7B, D). Two patients developed grade 1–2 graft-versus-host disease (GVHD) (Fig. 7D). All four patients experienced grade 2–4 neutropenia, monocytopenia, and thrombocytopenia (Fig. 7F). One patient developed sepsis, successfully treated with antibiotics.

The study has been halted due to safety concerns (DLT and severe infection), and the decision was made by the IRB based on "Termination Criteria" in the study protocol.

### Activity of C-JUN-overexpressing CD33 CAR T cells in patients with r/r AML

One patient underwent tumor burden assessment after lymphodepletion and before infusion, and lymphodepletion did not completely eliminate tumor cells (16.86% in BM). The remaining three patients refused to undergo a biopsy after lymphodepletion. Two patients with high baseline tumor burden achieved minimal residual disease-negative (MRD⁻) CR with incomplete hematologic recovery (CRi) on day 15, which was maintained until day 30 (Fig. 8A, B). Patient 4 achieved MRD⁺ CRi on day 15 but did not achieve CR on day 30 (Fig. 8A, B). Patient 1 achieved partial remission (PR) on day 15, but disease rebounded and was assessed as no response (NR) on day 30. This patient underwent a second infusion and achieved MRD⁺ CRi after 15 days (Fig. 8A, B). We also observed myelosuppression after CD33 CAR T cell therapy (Supplementary Fig. 9A). The CR/CRi patients underwent SCT at 35, and 38 days after infusion or at 29 days after the second infusion, respectively (Fig. 8A). By cutoff, two patients remained disease-free and alive for more than 2 years. One patient experienced a transient CD33⁺ disease relapse after 1 year, the patient was subsequently treated with Selinexor and Artesunate, reached remission, and underwent transplantation (Supplementary Fig. 9B).

CAR T cells expanded significantly in all patients. The two single-dose CRi patients had peak transgene levels of 5744 and 3966 copies/μg, respectively (Fig. 8C). The NR patient had a relatively low peak transgene level (109 copies/μg). The other patient had a peak transgene level of 4999 copies/μg after the first infusion and a peak level of 11,350 copies/μg after the second infusion (Fig. 8C). The two single-dose CRi patients had peak CAR T-cell counts of 192 and 23.5 cells/μl, respectively. The NR patient had a peak count of 10.3 cells/μl. The twice-infusion patient achieved a peak count of 2.12 and 1.73 cells/μl after the first and second infusion, respectively (Fig. 8D). In addition, the single-dose CRi patients had a higher proportion of CAR T cells (46.8% and 53.0%) among lymphocytes (Fig. 8E). As expected, peripheral CD33⁺ myeloid cells decreased (Fig. 8F).

### Discussion

In this study, we demonstrated that CAR T cells exposed to AML cells had impaired activation of calcium, ZAP70, ERK, and C-JUN. C-JUN is

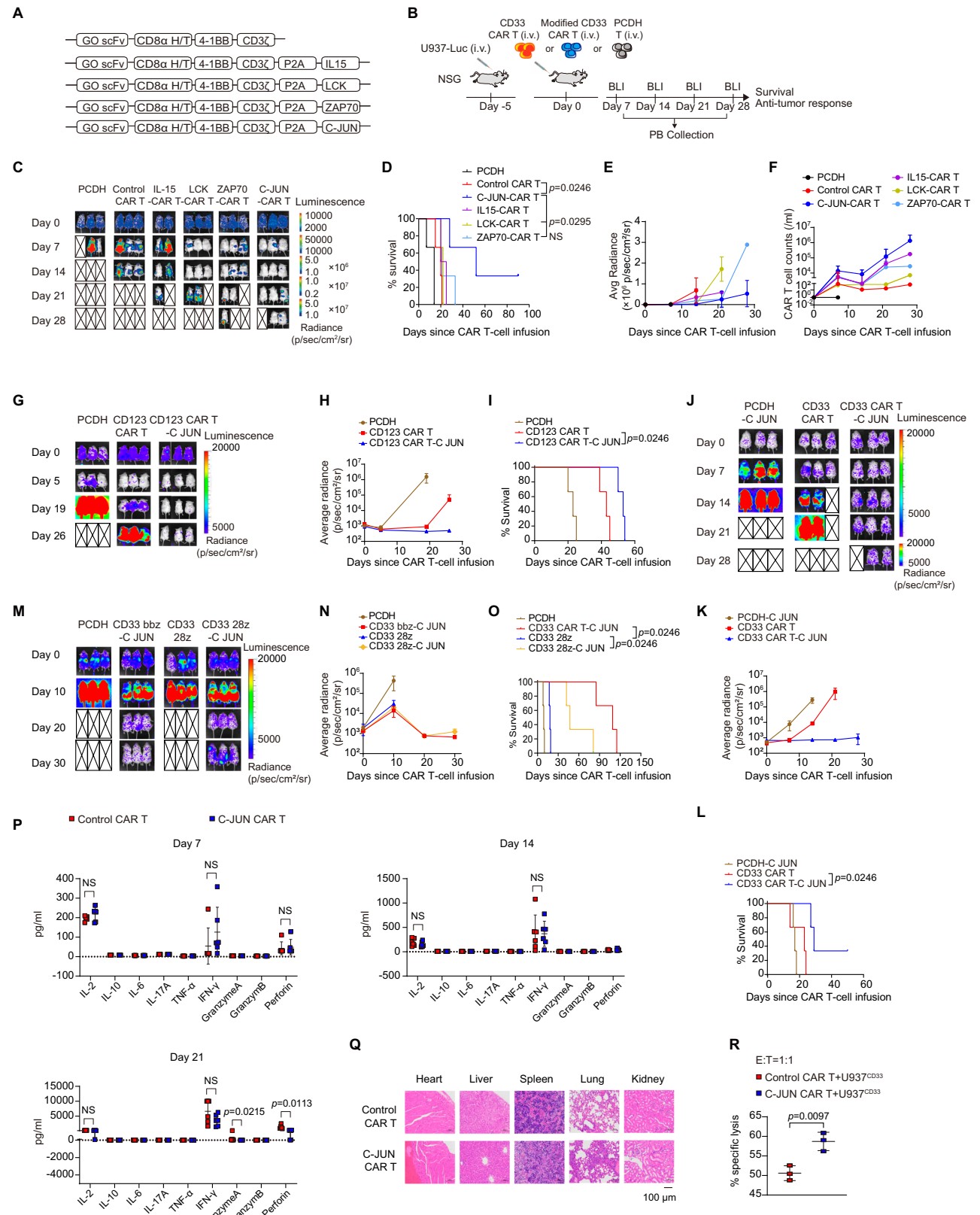

regulated by both JNK and ERK1/2[39,40]. Since JNK activation remained unchanged, the decreased C-JUN activation may be due to impaired ERK activation. Previous studies have shown endogenous T-cell exhaustion in AML patients[41,42]. Although in exhausted T cells, PD1 may provide an inhibitory signal to repress ZAP70 and ERK activity[43], we did not observe increased expression of exhaustion markers such

as PD1, TIM3, and LAG3 in AML-exposed CAR T cells during the limited observation period. Whether prolonged exposure to AML can lead to exhaustion of CAR T cells needs to be further evaluated.

The defect of ERK activation in AML-exposed CAR T cells may be related to the decreased expression of previously reported upstream activators of ERK, such as *PDGFA*[15], *NTRK1*[16], *MTURN*[17], *IGF1*[18], and

**Fig. 4 | Overexpression of C-JUN restored anti-AML activity of CAR T cells.**
**A** Schematic of CAR constructs (gemtuzumab ozogamicin, GO). **B** Schematic of
mouse model. NSG mice received U937 cells intravenously followed by CAR T cells
or control PCDH T cells 5 days later. **C** Representative bioluminescence imaging of
mouse model after CAR T treatment, $n = 2$ in IL15-CAR T group, $n = 3$ in other
groups. Survival curve (**D**) and quantification of tumor burden (**E**) as indicated by
average radiance (p/sec/cm²/sr) of (**C**), $n = 2$ in IL15-CAR T group, $n = 3$ in other
groups. **F** CAR T-cell counts in PB, data are summarized from two independent
experiments, $n = 3$ in PCDH and IL15-CAR T group, $n = 4$ in LCK-CAR T and ZAP70-
CAR T group, $n = 5$ in Control CAR T and C-JUN-CAR T group. **G, J, M** Representative
bioluminescence imaging of mouse model after CAR T treatment, $n = 3$.
**H, K, N** Quantification of (**G, J, M**) showing the tumor burden as indicated by
average radiance (p/sec/cm²/sr), $n = 3$. (**I, L, O**) Survival curve of NSG mice in

(**G, J, M**), $n = 3$. **P** Cytokine levels in tail blood collected on days 7, 14, and 21 post-
infusion, $n = 6$. **Q** Pathological analysis of the heart, liver, spleen, lung, and kidney of
the representative two mice in the control and C-JUN CAR T-treatment group at
their terminal stage by using HE staining. Magnification, 200×. **R** Cytolytic activity
of control and C-JUN CAR T cells against U937 cells in vitro, CAR T cells were sorted
from the mice spleen, $n = 3$. For all bar plots, data are shown as mean ± SD. Assays
were performed on day 10 after T-cell initial activation. Two-sided unpaired t-tests
or multiple two-sided unpaired t tests were used to assess significance in (**P, R**).
Survival curves were compared using the log-rank Mantel-Cox test in (**D, I, L, O**). All
numbers defined by "$n$" indicate the number of biological replicates with different
mice. Data are representative of two independent experiments. NS not significant.
Source data are provided in the Source Data file.

*CFLAR*[19], and increased *DUSP1*[20], a phosphatase that inactivates ERK.
However, the precise effects of these dysregulated genes on ERK
pathway remain to be determined.

Intriguingly, myeloid-derived suppressor cells (MDSCs) related
suppressive mediators ROS[23] and NO[24] did not significantly contribute
to the impaired function of CAR T in AML. CD155 and CD112 have been
reported to be highly expressed in AML cells, and associated with poor
prognosis[27]. Some researchers have reported that the blockade of
CD155 or CD112 can increase T-cell mediated lysis of AML cells[27,44], but
the effect of CD155 on CAR T-cell therapy has not been reported. Here,
we provide new insight by showing that CD155 but not CD112 is
expressed at higher levels in AML than in ALL, and this difference partly
accounted for the lower ERK activation and anti-tumor function
against AML than against ALL. Surprisingly, the effect of CD155 is not
through interacting with its known receptors CD226, CD96, and TIGIT,
although TIGIT has a higher affinity and delivers an inhibitory signal[45].
CD155 may interact with an as-yet unidentified receptor to exert sup-
pressive effects, which warrants further investigation. In addition, we
accept that there may be some other differentially expressed mole-
cules that also contribute to the suppressive mechanism, but are not
yet discovered in the present study.

It is surprising that C-JUN outperforms other factors (ZAP70, LCK,
IL-15, etc.) reported to promote CAR T function[31–33,46,47] and can sig-
nificantly promote CAR T therapeutic efficacy in AML disease model.
This may be due to tumor specificity, e.g., the mechanism of resistance
in different tumors varies, so only a specific intervention can restore
CAR T function. The effect of C-JUN was not restricted to a specific
target antigen and was also applied to another myeloid antigen CD123.
Overexpression of IL-15 in CAR T cells has been reported to promote
expansion and partially control tumor progression in AML, but it only
improves mice survival after using a TNF-α inhibitor[32]. In this study, IL-
15 overexpression had some slight beneficial anti-AML effect, but this
effect is not as strong as overexpression of ZAP70 or C-JUN. Notably,
we did not observe any toxicities associated with IL-15 overexpression,
possibly owing to the differences in the mouse models and target
antigens. Overexpression of both IL-15 and IL-21 has been shown to
further enhance anti-tumor effect[48]. However, this combination may
increase the risk of severe toxicity due to excessive cytokine produc-
tion and therefore were not tested in the present study. CAR T cells
overexpressing LCK and ZAP70 showed effects in improving anti-
leukemia function, but they were not as good as C-JUN overexpression.
This may be due to that intervention of this upstream signaling could
not fully compensate for the broad defect in antigen signaling. This
specific advantage of C-JUN may be due to a combination of positive
feedback compensation for defective ERK activity and direct C-JUN
transcriptional activation of costimulatory/cytokine genes. Notably,
overexpression of another AP1 component, C-Fos, had no beneficial
effect. This may be due to the different functional properties of C-JUN/
C-Fos or that C-JUN interacts with other partners to exert its effects.

One of our findings is that C-JUN enforced CAR-T function in AML
in a way different from that reported by previous studies with B-ALL,

solid tumors, and other tumor models[49–51]. Mackall et al. first found
that overexpression of C-JUN enhanced anti-tumor responses by
downregulating inhibitory receptors PD-1 and CTLA-4, and upregu-
lating pro-survival/memory factors IL-7R and TCF7 to resist
exhaustion[49]. However, we did not observe an obvious effect of C-JUN
overexpression in resisting CAR T exhaustion with regard to hallmark
inhibitory receptors and transcription factors. This discrepancy may
be due to several factors. First, the study by Crystal L. Mackall et al. was
specifically designed to address T-cell exhaustion and used disease
models, target antigens, or CAR costimulatory domains that are prone
to exhaustion, such as HA-28z and GD2-28z[52,53]. In contrast, no studies
have reported that CD33 CAR T is susceptible to exhaustion in AML,
and CD33-bbz, the main construct we used, was less susceptible to
exhaustion compared to CD33-28z CAR T cells[54]. In addition, the lim-
ited in vivo persistence time of control CD33 CAR T cells hindered our
ability to evaluate the long-term effects. Third, the lack of exhaustion
of CAR T in AML may be caused by the property of this tumor and the
resistance mechanisms. As shown in our study, the CAR T in AML
mainly manifested as hypoactivation and decreased effector program,
indicating an insufficient stimulatory signal. This specific defect of
CAR-T may lead to a different mode of action of C-JUN overexpression.

C-JUN overexpression increased the expression of co-stimulatory
molecules and cytokines in CAR-T with AML. It has been documented
that C-JUN induces the expression of IL-2, IFN-γ, and TNF-α[55]. In AML-
exposed CAR T cells, C-JUN did promote IL-2 expression at the RNA
and protein levels, whereas IFN-γ was only upregulated at the protein
level, suggesting that C-JUN may promote anti-tumor efficacy at least
partly by increasing these cytokines. However, C-JUN did not increase
TNF-α expression, which is critically involved in CRS in AML, and this
may pose a special advantage of C-JUN with regard to CAR T-cell
functional optimization in AML. It should be noted that C-JUN CAR
T cells did not increase chromatin accessibility for certain genes that
showed altered expression at the mRNA level. This discrepancy may
rely on that some effect of C-JUN is mediated by modulation of pro-
moter activity rather than changing chromatin accessibility.

CAR T cell efficacy may be influenced by different construct fea-
tures such as costimulatory molecules and single-chain variable frag-
ments (scFvs). We observed similar defects of CAR T in AML regardless
of the costimulatory domains (CD28 and 4-1BB) or antigen targets
(CD33 and CD123). We have not specifically investigated the different
scFvs of CD33. A previous study demonstrated that CAR T cells with
gemtuzumab scFv and 4-1BB had relatively lower efficacy in AML mice
than other scFvs, but the change to CD28 costimulatory domain
improved the efficacy[56], and the previous clinical trials on CD33 CAR
T cells with different scFvs uniformly showed incapability to induce
disease remission and little expansion[7,8]. Therefore, overexpression of
C-JUN is worth testing and may have similar beneficial effects in CAR
T cells with different anti-CD33 scFvs.

Many findings from the preliminary trial may provide useful
information for future therapy. Previous trials of CD33 CAR T-cell
therapy in AML cannot be compared to our results due to the

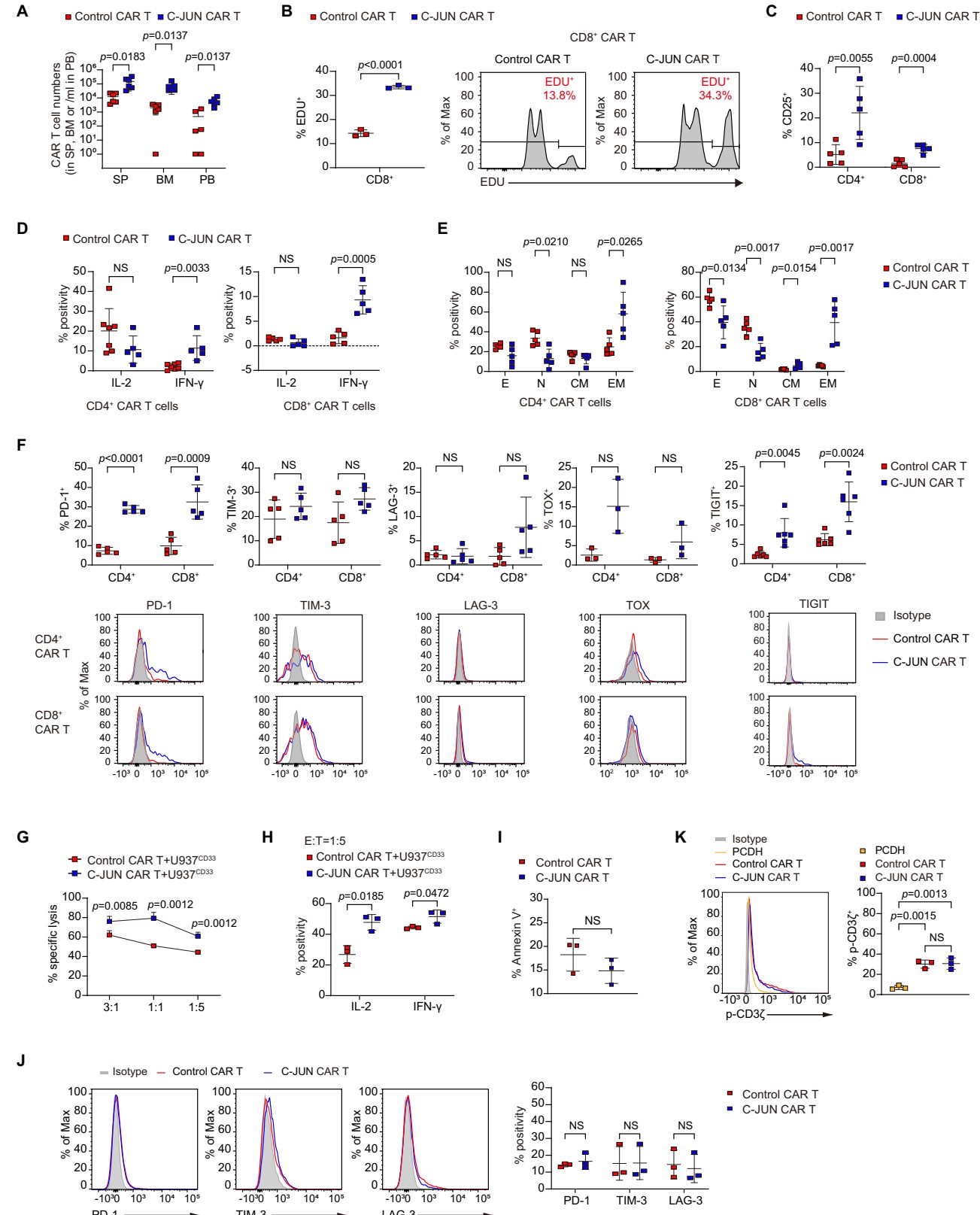

differences in patient characteristics and treatment procedures. In our trial, severe CRS was observed in one patient, who recovered after anti-TNF therapy, highlighting the value of TNF blockade in future therapy. Importantly, there was no severe neurotoxicity. During CD19 CAR T-cell therapy, IL-6 levels increased dramatically and correlated with severity of CRS[57]. In contrast, we observed a greater increase of TNF-α than other cytokines, which may provide evidence for the effectiveness of anti-TNF-α therapy in managing CRS in AML. All patients in remission were bridged to SCT, as done by others[58], which promoted myeloid recovery and may have reduced the long-term risk of infection. Of the three patients achieving remission, no CD33 loss relapse occurred, suggesting that CD33 may be a stable target.

**Fig. 5 | The effect of C-JUN overexpression in the phenotype of CAR T cells.**
**A.** CAR T cell counts in SP, BM, and PB collected from day 10 to 20 post-infusion, $n = 6$. **B** Percentage of EDU$^+$ cells in CD8$^+$ CAR T cells, $n = 3$. **C** Percentage of CD25$^+$ cells in CAR T cells, $n = 5$. **D** Percentage of IL-2 and IFN-γ in CAR T cells, $n = 7$ in CD4$^+$ CAR T cells in the control CAR T group, $n = 5$ otherwise. **E** Percentage of naïve (N, CD45RA$^+$CD62L$^+$), central memory (CM, CD45RA$^-$CD62L$^+$), effector memory (EM, CD45RA$^-$CD62L$^-$), and effector (E, CD45RA$^+$CD62L$^-$) cells in CAR T cells, $n = 5$.
**F** Percentage of PD-1, TIM-3, LAG-3, TOX, and TIGIT in CAR T cells, $n = 3$ in TOX, $n = 6$ in TIGIT, $n = 5$ otherwise. **G** Cytolytic activity of control and C-JUN CAR T cells against U937 cells in vitro, CAR T cells were cocultured with U937 cells at the indicated E:T ratios, $n = 3$ replicates per point, representative of three donors.
**H** Percentage of IL-2 and IFN-γ in CAR T cells cocultured with U937 cells in vitro, $n = 3$. **I** Percentage of Annexin V$^+$ in control and C-JUN CAR T cells cultured alone, $n = 3$. **J** Percentage of PD-1, TIM-3, and LAG-3 in control and C-JUN CAR T cells cultured alone, $n = 3$. **K** Expression of phosphorylated CD3ζ in control CAR-T, C-JUN CAR-T, and PCDH (non-CAR transduction) T cells cultured alone, $n = 3$. For all bar plots, data are shown as mean ± SD. Assays were performed on day 10 after T-cell initial activation. Two-sided unpaired $t$-tests or multiple two-sided unpaired $t$ tests were used to assess significance in (**A**–**J**). One-way ANOVA was used in (**K**). All numbers defined by "$n$" indicate the number of biological replicates with different human donors or mice. Data are representative of two independent experiments. NS not significant. Source data are provided in the Source Data file.

Some limitations of this study should be considered. First, because of the small sample size, early halting of the study, and single-arm design, it is premature to conclude that C-JUN-overexpressing CAR T cells enhanced efficacy than regular CAR T cells. Second, the adverse events of the approach need attention. A major concern was that the second patient experienced a grade 4 CRS. Although anti-TNFα has been shown to effectively treat CRS, infections remain a significant risk. Infections may not reflect any flaw in CAR T products. Rather, they were due to the expression of CD33 on normal myeloid cells. We are considering necessary adjustments in future research, such as adjusting lymphodepletion regimens, adjusting the timing of SCT, and other infection-preventing approaches. There is also some room for further improvement of optimized CAR T cells to increase durability. However, it is important to note that excessive expansion of CAR T cells may also increase the risk.

Taken together, these findings demonstrate that AML suppress antigen receptor signaling in CAR T cells to restrain anti-leukemia function. CD155 is partially responsible for this inhibition. Overexpression of C-JUN has the potential to restore effective anti-AML response. Optimized CD33 CAR T cells show preliminary activity in the clinic, generating several signs that support the further development of CAR T cell therapy for AML. More broadly, in addition to the well-established exhaustion model, T-cell hypoactivation and insufficient antigen signaling deserve attention in future CAR T therapy studies with different types of tumors.

## Methods

### Study approval and ethics statements
All experiments involving human samples were approved by the Ethics Committee of the Institute of Hematology and Blood Diseases Hospital (approval number: NKRDP2021009-EC-2). All animal experiments performed in this study were in strict accordance with institutional guidelines and were approved by the Institutional Animal Care and Use Committees of State Key Laboratory of Experimental Hematology (SKLEH). The mice were housed in a specific pathogen-free (SPF) environment.

The Phase I, open-label study (NCT04835519) was conducted at Beijing GoBroad Boren Hospital. The study protocol was approved by the Institutional Review Board (IRB) of Beijing GoBroad Boren Hospital. Written informed consent was obtained from the patients or their legal guardians before enrollment (ethics approval number: 20210331-TY-001K). All clinical investigations were conducted in accordance with the Declaration of Helsinki principles. The study protocol is available in the Supplementary Information file.

### Generation of CAR constructs
This study generated CAR constructs by combining scFvs that are specific for human CD33 (clone My96), CD123 (clone 32716), CLL1 (clone 1075.7), CD19 (clone FMC63), or CD38 (clone 056) with a CD8α hinge-transmembrane (H/T) domain, a 4-1BB costimulatory endodomain, or CD28 H/T domain, a CD28 costimulatory endodomain, and the CD3ζ intracellular domain. They were cloned into the pCDH-EF1-MCS-Puro lentiviral vector. CAR fragment was linked to IL-15, C-JUN,

ZAP70, LCK, C-Fos, or IFN-γ cDNA sequences by a P2A self-cleaving peptide. "PCDH" T cells refer to T cells transduced with an empty pCDH vector, which serves as a negative control.

### Cell lines
Cell lines with strict quality control were used in the study. Specifically, short tandem repeat fingerprinting was performed within the last year, and PCR validation was done to guarantee that the cell lines were not contaminated with mycoplasma. The human cell lines were derived from our institute's cell resource center in the study. Green fluorescent protein (GFP) and firefly luciferase were stably transduced into some cell lines using lentivirus. The CRISPR-Cas9 method was used to conduct CD155, CD33, and CLL1 gene knockout in U937 tumor cells. CD96 and TIGIT gene knockout were achieved in CD33 CAR T cells using the CRISPR-Cas9 system. Nalm6 and U937 were transduced with lentiviral vectors that encoded CD33, CLL1, or CD123. A lentiviral vector encoding CD155 was used to transduce Nalm6. Knockout or over-expressed cell lines were further purified via single-cell sorting or multiple rounds of flow sorting (FACS Aria II, BD Biosciences), and then were tested by PCR to ascertain mycoplasma contamination-free. Nalm6, Raji, U937, THP-1, and HL60 cell lines are cultured in RPMI-1640 (Gibco). SUP-B15 is grown using IMDM (Gibco). DMEM medium (Gibco) was used to culture HEK293T cells.

The Beijing Gobroad Boren Hospital's Hematology Department and our Institute's Clinical Laboratory provided the primary AML and ALL samples. Gradient centrifugation was used to isolate human T cells from healthy individuals' peripheral blood. Our institute approved all procedures (approval number: NKRDP2021009-EC-2). We cultured T cells in ImmunoCult™-XF T Cell Expansion Medium (Stem Cell Technologies), which included recombinant human IL-2 (Peprotech).

### Lentiviral vector production
In the preclinical study, third-generation self-inactivating CAR-encoding lentiviral supernatants were generated using the HEK293T packaging cell line. Polyethyleneimine (PEI, Polysciences) was used as the transfection agent to co-transfect 80% confluent HEK293T 10 cm dishes with the target vector plasmid, Rev, pMDL, and VSVG packaging plasmid DNA. After 2 days, the virus-containing supernatant was harvested and concentrated. Concentrated lentivirus stocks were kept at −80 °C. A clinical grade CD33 CAR lentiviral vector was produced at Beijing Gobroad Boren Hospital per current good manufacturing practice standards.

### CAR T cell manufacture
By following the reagent's instructions, the EasySepTM Human T cell Enrichment Kit was applied to isolate human primary T cells, and the ImmunoCult™ Human CD3/CD28 T Cell Activator was used to activate T cells, which were then transduced using a lentiviral vector 48 h after activation and grown in the T cell media containing IL-2. All reagents were from Stem Cell Technologies. If not stated elsewhere, on days 10−11 following initial T cell activation using CD3/CD28 T cell activator, we used CAR T cells in vitro experiments or implanted them into mouse models[49].

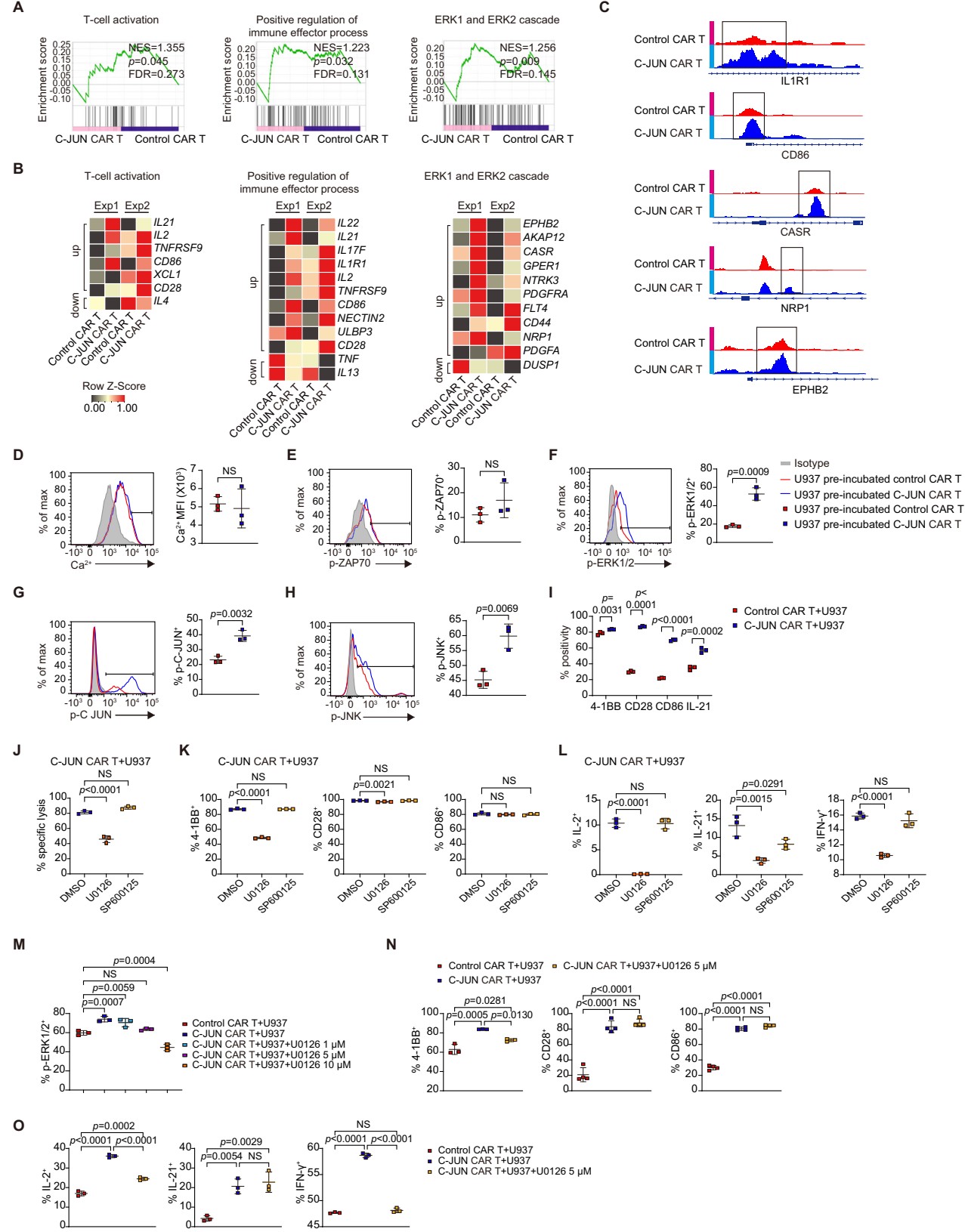

## CRISPR knockout

CRISPR/Cas9 gene editing was performed by electroporation of Cas9/ guide RNA (gRNA) ribonucleoprotein (RNP) complex using P3 Primary Cell 4D-Nucleofector™ X Kit S (Lonza). 10 μg Cas9 protein (Invitrogen) and 100 pmol chemically modified gRNA (Ubigene) in each reaction were pre-compounded for 45 min at 25 °C to form RNP complexes.

Cells were suspended in RNP transfection buffer at $1.5–2 \times 10^6$ cells per 20 μl reaction and were electroporated in 16-well cuvette strips per the protocol of the manufacturer. After electroporation, cells were pipetted out and resuspended in a pre-warmed Medium and expanded. The efficiency of knockdown was then evaluated by flow cytometry after 48 h. The non-targeting sgRNAs were electroporated into control cells.

**Fig. 6 | C-JUN re-activates ERK and upregulates co-stimulatory molecules and cytokines. A, B** GSEA results from RNAseq of U937 co-incubated- control versus C-JUN CAR T (**A**). Nominal *P* values, FDR *q* values, and NES were calculated using GSEA software (Broad Institute). Heat maps (**B**) indicating the expression of genes enriched in GSEA from (**A**) and the known related genes not in the GSEA gene set. The genes in heatmaps meet: fold change ≥ 1.5-fold in each of the two biological replicates. Each RNA sample was pooled from three technical replicates with T cells from one donor, and we conducted experiment with two different donors, *n* = 2. **C** Differentially accessible regions of indicated genes from ATAC-seq analysis. **D** Intensity of intracellular calcium in CAR T cells, *n* = 3. Expression of p-ZAP70 (**E**), p-ERK1/2 (**F**), p-C-JUN (**G**), and p-JNK (**H**) in CAR T cells pre-incubated with U937 and re-stimulated with U937, *n* = 3. **I** Percentage of 4-1BB, CD28, CD86, and IL-21 in CAR T cells cocultured with U937, *n* = 3. **J** Cytolytic activity of C-JUN CAR T cells against

U937 with 50 μM U0126 or 50 μM SP600125, *n* = 3. Percentage of 4-1BB, CD28, CD86 (**K**), and IL-2, IL-21, IFN-γ (**L**) in T cells from (**J**), *n* = 3. (**M**) Percentage of p-ERK1/2 in U937 pre-incubated-control CAR T, C-JUN CAR T, and C-JUN CAR T with 1, 5, 10 μM U0126, *n* = 3. Percentage of 4-1BB, CD28, and CD86 (**N**) and IL-2, IL-21, and IFN-γ (**O**) in U937-co-incubated control CAR T, C-JUN CAR T, and C-JUN CAR T with 5 μM U0126, *n* = 4 in CD28 and CD86 expression, *n* = 3 otherwise. For all bar plots, data are shown as mean ± SD. Assays were performed on day 10 after T-cell initial activation. Two-sided unpaired *t*-tests or multiple two-sided unpaired *t* tests were used to assess significance in (**D–I**). One-way ANOVA was used in (**J–O**). All numbers defined by "*n*" indicate the number of biological replicates with different human donors. Data are representative of two independent experiments. NS not significant. Source data are provided in the Source Data file.

The following sgRNA targeting sequences were used: CD33[59], 5'-GTC AGTGACGGTACAGGA-3', CLL-1, 5'-CGCCATACATGAGAGGGAGC-3', CD155, 5'-CAAGCCCCAGAACACAGCTGAGG-3', CD96[60], 5'-CGTGCA GATGCAATGGTCCA-3', TIGIT[60], 5'-TCCTCCTGATCTGGGCCCAG-3'.

## Flow cytometry

The samples were processed using LSR Fortessa or CantoII (BD Bioscience), and result analysis was done with FlowJo 9.0 or 10 software. CD33, CD123, CLL1, CD19, and CD38 CAR expression were measured using human CD33-Fc, CD123-His, CLL1-Fc, CD19-His and CD38-His proteins (Acro) and anti-human IgG Fc or anti-His tag antibodies (BioLegend). The expression of C-JUN, LCK, and ZAP70 was detected by intracellular staining using the following antibodies: anti-LCK (Biolegend, clone: LCK-01), anti-ZAP70 (Biolegend, clone: A15114B), and anti-C-JUN (Cell Signaling Technology, clone: 60A8). The supplementary information included information about other antibodies employed in the study.

Countbright absolute counting beads (BioLegend) were used to calculate the absolute cell counts obtained by flow cytometry. We tested cell viability by DAPI (Solarbio), 7-AAD (BD Bioscience), or Fixable Viability Dye (FVD, eBioscience).

## Intracellular cytokine staining

Following the specified effector-to-target ratios for coculturing CAR+ T and tumor cells, PMA/ionomycin mixture (250×) (Multi Sciencess) and GolgiStop (BD Bioscience) were added and incubated for 5.5 h. Incubation with the appropriate fluorescence-labeled antibodies was used to stain the surface proteins. Cell viability was determined by FVD (eBioscience). We measured the levels of intracellular cytokines with the FoxP3 Staining Buffer Set (eBioscience).

## Phosphor-specific flow cytometry

We followed the instructions on the reagent to conduct the phosphor-specific flow cytometry. We co-incubated CAR+ T with tumor cells for 12 h and then employed fluorescence-activated cell sorting (FACS) to sort CAR+ T cells to allow them some time away from the leukemia cells so that they can be more sensitive to the second stimulation with regard to phosphorylation events. We re-stimulated these sorted CAR T cells with U937 cells, CD33-knockout U937 cells, PMA (Multi Sciences), and anti-CD3 (Cell Signaling Technology) for the indicated time (15–30 min) at 37 °C to observe the phosphorylation of signaling molecules, and the cells were incubated with FVD (eBioscience) as a live/dead marker. We next used 4% paraformaldehyde to fix CAR T cells for 20 min at 25 °C. After washing with staining buffer (PBS containing 1% BSA), the cells permeabilized for 30 min on ice by adding a final concentration of 90% (v/v) ice-cold methanol. After extensive washing, cells were stained at 25 °C for 60–120 min with the following antibodies: PE Cy7 phos-ERK1/2 (Thr202/Tyr204, BioLegend), phos-ZAP70 (Tyr319, Cell Signaling Technologies), phos-JNK (Thr183/Tyr185, Cell Signaling Technologies), and phos-C-JUN (Ser73, Cell Signaling Technologies). For the detection of phosphorylated ZAP70,

JNK, and C-JUN, anti-rabbit IgG conjugated to PE (TransGen Biotech) was used as the secondary antibody, and rabbit mAb IgG was used as the control (Cell Signaling Technologies).

## Intracellular calcium concentrations

Following the description of the reagent, the Fluo-4 AM Calcium Indicator (Beyotime) was employed to measure the intracellular calcium concentrations. The fluorescence intensity was determined by LSR Fortessa (BD Biosciences).

## Measurement of intracellular ROS and NO

We measured intracellular levels of ROS and NO with the Reactive Oxygen Species Assay Kit and DAF-FM DA (Beyotime). Cells were plated and measured according to the manufacturer's recommendations.

## In vitro cytotoxicity assays

In brief, in the 96-well U-bottom plate, target cells were cocultured with CAR T or PCDH T cells for 12 h without the addition of IL-2. Control experiments with only target cells were performed to adjust the position of live tumor cells. To make them distinguishable from the tumor cells, the T cells were labeled using a CD3 antibody. CAR T cell cytotoxicity was assessed by calculating the proportion and number of surviving target cells. When indicated, we added ROS inhibitor 2-ME (55 μM, Thermo), NO inhibitor L-NMMA (10 μM, MCE), anti-CD155 (500 ng/ml, Biolegend), anti-CD112 (10 μg/ml, Biolegend), anti-TIGIT (50 μg/ml, Biolegend), anti-CD96 (20 μg/ml, Biolegend), anti-CD226 (10 μg/ml, Biolegend), JNK inhibitor SP600125 (50 μM, MCE), and ERK1/2 inhibitor U0126 (1, 5, 10, 50 μM, MCE) to the coculture system[61–63]. The concentrations stated in this study were either obtained via concentration titration or selected based on the manufacturer's instructions or published data. The final used concentration was not toxic to cells.

For in vitro cytotoxicity assays against primary samples, we first identified tumor cells using the CD45/SSC gate and then used AML/ALL-specific markers (e.g., CD34, CD117, CD33, CD38, CD19, and CD22) to distinguish blast cells from normal cells. We then sorted primary AML and ALL samples expressing the same CD38 antigen. In the in vitro killing assays, the remaining primary AML and ALL cells were determined by the CD3⁻CD38⁺ circle gate.

## Apoptosis assay

To detect apoptosis, we used the Annexin V-APC and 7-AAD/DAPI Apoptosis Detection Kit (Simubiotech). After collection, we stained the cells with 5 μl of Annexin V and 5 μl of 7-AAD/DAPI. Then we measured the samples by flow cytometry immediately.

## EdU proliferation assay

We measured CAR T cell proliferation using the EdU Cell Proliferation Kit (Beyotime). Mice were given an intraperitoneal injection of EdU (10 mg/kg) for labeling. Twelve hours later, the tail blood samples were collected from NSG mice and measured per the instructions. Data were collected using a Canto II instrument (BD Biosciences).

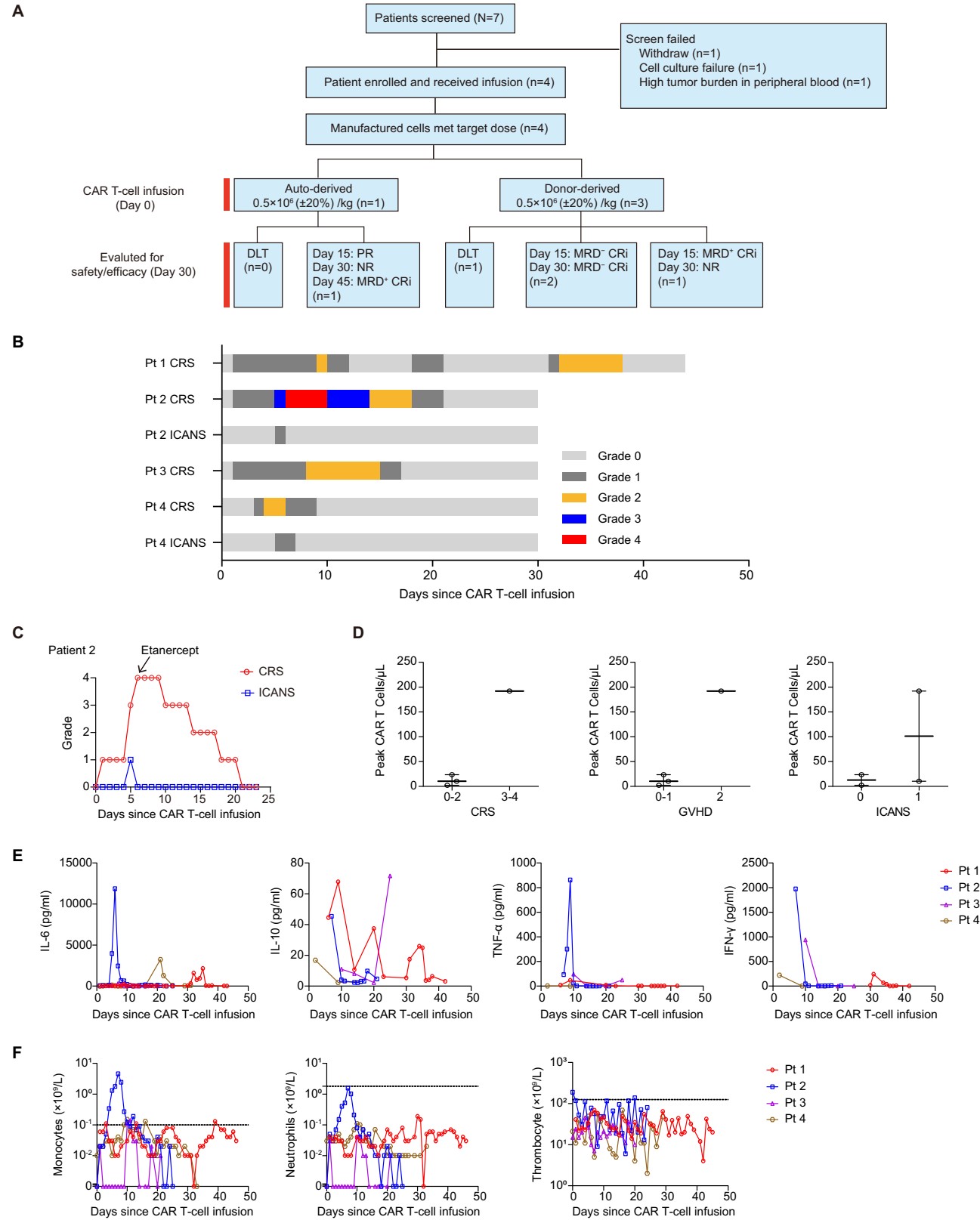

**Fig. 7 | Safety of C-JUN-overexpressing CD33 CAR T cells in patients with r/r AML. A** CONSORT diagram of the clinical trial. **B** Swimmer plot (*n* = 4) demonstrating the occurrence of CRS and ICANS after the infusion of CAR T cells. Each bar represents an individual patient. The severity is indicated by different colors. **C** CRS and ICANS management with etanercept in patient 2, *n* = 1. **D** The peak numbers of CAR T cells in PB of patients according to the grade of CRS, GVHD, and ICANS. Each dot represents one patient, *n* = 4. **E** Kinetics of serum cytokines for all patients in the first 30 days after CAR T-cell infusion. Each line represents one patient, *n* = 4. **F** Kinetics of monocyte, neutrophil, and thrombocyte counts in PB at different time points post-infusion. Each line represents one patient, *n* = 4. CRS cytokine release syndrome, ICANS immune effector cell-associated neurotoxicity syndrome, GVHD Graft-versus-Host Disease. Source data are provided in the Source Data file.

**Table 1 | Patient baseline characteristics and clinical outcomes**

| Pt (no.) | Prior Lines of Therapies (no.) | Prior SCT | Disease Status — Disease Burden | | | | Clinical Outcomes — Infusion Dose ($10^6$/kg) | Safety | | | Efficacy | | |
|---|---|---|---|---|---|---|---|---|---|---|---|---|---|
| | | | BM blasts (%, by morphology) | BM blasts (%, by FCM) | EMD | CD33 Expression (%, by FCM) | | CRS | Neurotoxicity | DLT | Response at day 15 | Response at day 30 | Response within 3 months |
| 1 | 2 | NO | 22.50 | 21.05 | NO | 99.79 | 0.5 | 1 | 0 | 0 | PR | NR | MRD$^+$ CRi |
| 2 | 4 | Allo-SCT | 86.50 | 61.96 | NO | 99.93 | 0.5 | 4 | 1 | 1 | MRD$^-$ CRi | MRD$^-$ CRi | MRD$^-$ CRi |
| 3 | 6 | Allo-SCT | 67.00 | 51.11 | NO | 98.80 | 0.5 | 2 | 0 | 0 | MRD$^-$ CRi | MRD$^-$ CRi | MRD$^-$ CRi |
| 4 | 4 | Allo-SCT | 10.50 | 4.35 | NO | 99.30 | 0.5 | 2 | 1 | 0 | MRD$^+$ CRi | NR | NR |

AML acute myeloid leukemia, BM bone marrow, CRi complete remission with incomplete hematologic recovery, CRS cytokine release syndrome, DLT dose-limiting toxicity, EMD extramedullary disease, MRD minimal residual disease, NR no response, PR partial remission, Pt patient, SCT stem cell transplantation.
Patient 1 received a second CAR T-cell infusion ($3.76 \times 10^6$/kg) on day 29 after the first CAR T-cell infusion. During the second infusion, the patient developed grade 2 CRS and was assessed as MRD$^-$ CRi 15 days after the second infusion, and the patient subsequently received SCT and has maintained disease-free.

### Enzyme-Linked Immunosorbent Assay
The QuantiCyto® Human IL-15 ELISA Kit (NEOBIOSCIENCE) was used to quantify IL-15 expression as recommended. Data were acquired on a Synergy H4 Hybrid Enzyme Labeler (BioTek). The sample concentration was determined from a standard curve and the sensitivity of the kit was 7.8 pg/ml.

### Cytokine measurements
The LEGENDplex™ Human CD8/NK Panel (13-plex) (Biolegend) was used to measure cytokines in tail blood. Data was analyzed on the online platform (https://legendplex.qognit.com).

### Histological analysis
Mice were euthanized for histological examination. Hematoxylin-eosin (HE) was used to stain the heart, liver, spleen, lung, and kidney when they had been preserved in 4% paraformaldehyde.

### Mouse experiments
This study was approved by the Institutional Animal Care and Use Committees of SKLEH. We followed the institutional guidelines for the care and use of laboratory animals when conducting all experimental procedures. We conducted mouse experiments with female NOD.Cg-$Prkdc^{scid}Il2rg^{tm1Wjl}$/SzJ (NSG) mice aged 6- to 10 weeks from Beijing Biocytogen and maintained under conditions free of pathogens. To explore the difference between treating mice bearing myeloid and B-lineage tumors with CAR T cells, we injected CAR T cells and tumor cells at various ratios. A model with identical tumor burdens was established when CAR T cells were infused. Mice were tail-vein injected with $(0.3-1) \times 10^6$ Nalm6$^{CD33}$ or U937$^{CD33}$ cells that were engineered for luciferase expression (Nalm6$^{CD33}$-Luc or U937$^{CD33}$-Luc), followed by $1 \times 10^6$ CAR T cells or PCDH T cells 3–5 days later.

To monitor tumor burden, we intraperitoneally injected the mice using D-Luciferin potassium salt (YEASEN) and imaged them with IVIS at indicated time points. Living Image software (PerkinElmer) was used for visualization and calculation of average luminescence. Two to three mice were used for different experimental conditions, and the experiments were independently replicated two to three times. We randomly assigned the mice to receive CAR T cell therapy after tumor cell implantation. We made a decision to euthanize the mice based on the level of discomfort observed in the mice, specifically including symptoms such as a hunched posture, significant weight loss, mobility problems, or paralysis.

### RNA sequencing and GSEA
The Illumina NovaSeq 6000 platform (BerryGenomics, China, and Novogene, China) was used for the RNA sequencing. After 2 days of exposure to the indicated target cells, CD8$^+$ CAR$^+$ T cells were sorted by FACS for bulk RNA-seq. The sequencing was done using the 150 bp paired-end setting. The DeSeq2 R package was applied to determine differentially expressed genes for two conditions[64]. We generated heatmaps using TBtools (https://github.com/CJ-Chen/Tbtools). The heatmaps are based on log$_2$-transformed expression values. Row z-scores are displayed and were calculated using TBtools. We conducted GSEA enrichment analysis with the GSEA software (Broad Institute) as described[65]. We used the GSEA software to calculate the false discovery rate and normalized enrichment score for the RNAseq data. The T-cell exhaustion gene set was from the previously described data[66]. The Broad Institute Molecular Signature Database provided the other gene sets (GSEA|MsigDB (gsea-msigdb.org)) as follows:

 GOBP_ALPHA_BETA_T_CELL_ACTIVATION
 PID_TCR_CALCIUM_PATHWAY
 GOBP_CYTOKINE_PRODUCTION
 ST_JNK_MAPK_PATHWAY
 GOBP_POSITIVE_R-
EGULATION_OF_IMMUNE_EFFECTOR_PROCESS

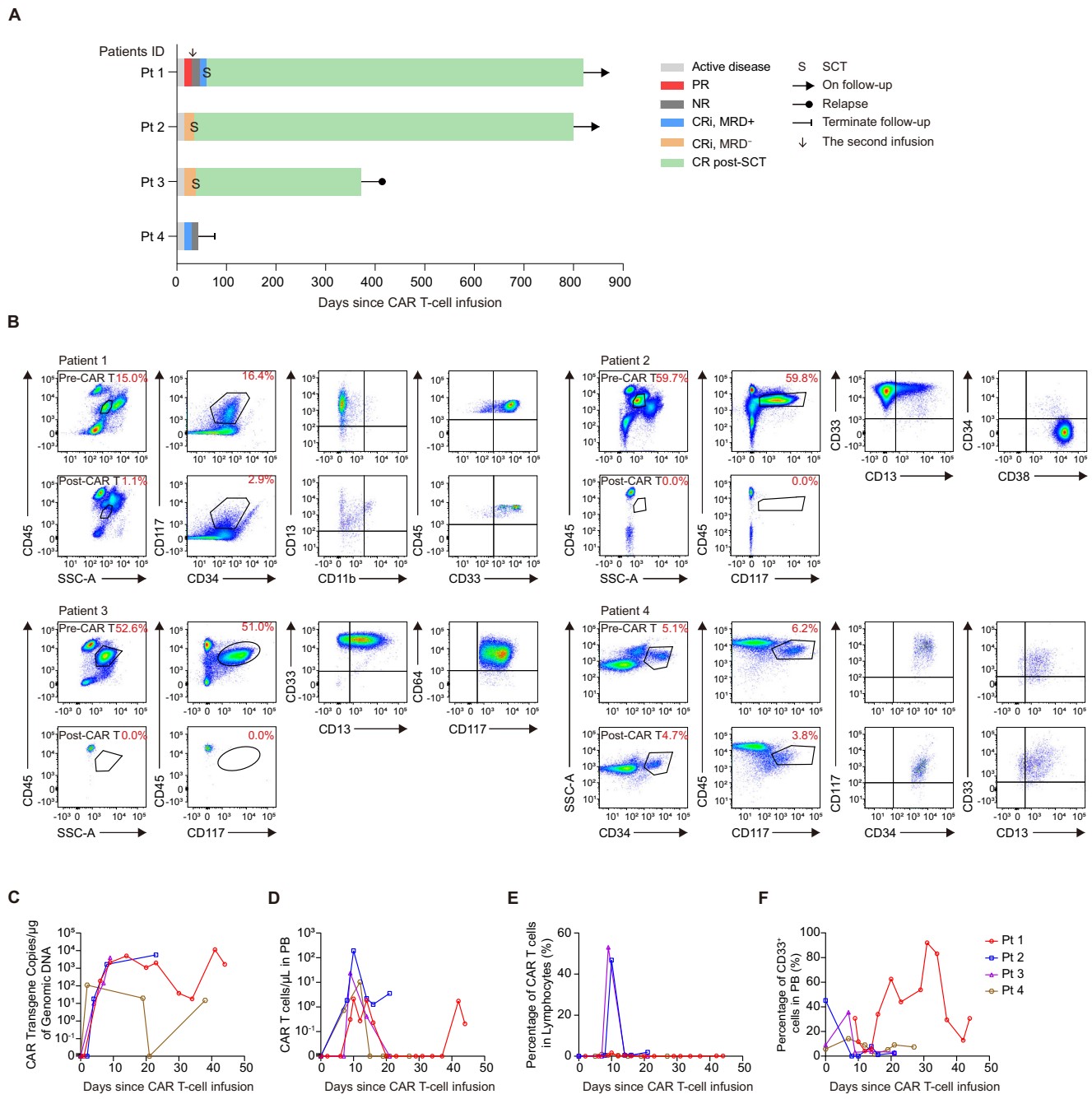

**Fig. 8 | Activity of C-JUN-overexpressing CD33 CAR T cells in patients with r/r AML. A** Swimmer plot (*n* = 4) showing patient responses. Each bar represents an individual patient. Responses were determined on day 15 and day 30 and were indicated by different colors. Bars with solid arrows represent patients in an ongoing follow-up. **B** Dot plots indicating blasts in the BM samples before CAR T-cell infusion (patient 1 at 21 days, patient 2 at 72 days, patient 3 at 12 days, patient 4 at 12 days) and on day 30 post-infusion (except patient 1 who was analyzed on day 15 post the second CAR T-cell infusion), as determined by flow cytometry. Kinetics

of CAR vector transgene copies (**C**), CAR T cell counts (**D**), percentage of CAR T cells on lymphocytes (**E**), and percentage of CD33⁺ cells in PB (**F**) of individual patients at different time points post-infusion, as determined by quantitative PCR and flow cytometry. Each line represents one patient, *n* = 4. CR complete remission, CRi complete remission with incomplete hematologic recovery, DLT dose-limiting toxicity, MRD minimal residual disease, NR no response, PR partial remission, Pt patient, SCT stem cell transplantation. Source data are provided in the Source Data file.

BIOCARTA_NFAT_PATHWAY
GOBP_ERK1_AND_ERK2_CASCADE

## ATAC-seq and analysis
We used high throughput sequencing (ATAC-seq) to perform the Assay for Transposase-Accessible Chromatin, referring to previous reports[67]. At least 150,000 live cells from every sample were obtained by FACS.

An Illumina HiSeq platform was used for the sequencing of the library preparations. Then paired-end readings of 150 bp were generated.

## Clinical trial design, patient treatment, and response assessments
The first-in-human, single-center, open-label, single-arm Phase 1 clinical trial of functionally optimized CD33 CAR T-cell therapy for

patients with r/r AML was conducted at Beijing GoBroad Boren Hospital. The trial was registered with ClinicalTrials.gov (identifier NCT04835519, date April 5, 2021). The protocol can be accessed in the supplementary information files and the IRB of Beijing GoBroad Boren Hospital approved the protocol. Prior to enrollment, patients or their guardians gave written informed consent in compliance with the Declaration of Helsinki's ethical guidelines (the ethics approval number: 20210331-TY-001K). Currently, the study has been finally suspended due to safety concerns (DLT and severe infection), and the decision was made by the IRB based on "Termination Criteria" in the study protocol. A new trial may be initiated after the protocol and treatment procedure are adjusted to improve safety. As our clinical trial is currently suspended, we have received IRB approval to report data on these 4 patients.

These four patients were enrolled following the 3 + 3 design. The primary endpoint was safety. The secondary endpoint was efficacy. Patients received a lymphodepletion chemotherapy regimen consisting of fludarabine and cyclophosphamide. The four patients enrolled underwent CAR T cell infusion between April 13, 2021, and July 28, 2021. The infusion dose of CAR-T cells is $5 \times 10^5$ ($\pm20\%$) CAR T cells/kg. The data cut-off date was July 15, 2023. Notably, patient 1 was given a second CAR T infusion on day 29 due to poor clinical response and unsatisfactory expansion. The IRB has approved the change from "3 + 3" scheme to BOIN12 design after enrollment of the first four patients. However, no further patients were enrolled before the stop of the clinical trial.

Inclusion Criteria: To be eligible to participate in this study, an individual must meet all of the following criteria: 1. Candidates with relapse or refractory CD33$^+$ AML, who have progressed after treatment with all standard therapies or are intolerant of standard therapy, have limited prognosis with currently available therapies and had no available curative treatment options (such as SCT or chemotherapy); 2. Male or female, aged 1–70 years; 3. No serious allergic constitution; 4. Eastern Cooperative Oncology Group (ECOG) performance status score 0 to 2; 5. Have a life expectancy of at least 60 days based on the investigator's judgment; 6. CD33 positive in bone marrow or cerebrospinal fluid (CSF) by flow cytometry, or CD33 positive in tumor tissues by immunohistochemistry; (CD33 positive criteria: Flow cytometry: Positive: >80% of tumor cells expressed CD33 and the MFI of CD33 is the same as that in normal myeloid cells; Dim: >80% of tumor cells expressed CD33, but the MFI of CD33 is lower than that in normal myeloid cells as least as 1 log; Partial positive: 20–80% of tumor cells expressed CD33 and the MFI of CD33 is the same as that in normal myeloid cells. tumor tissue immunohistochemistry: Positive > 30% tumor cells expressed CD33); 7. Provide a signed informed consent before any screening procedure; subjects who voluntarily participate in the study should have the ability to understand and sign the informed consent form and be willing to follow the study visit schedule and relevant study procedure, as specified in the protocol. Candidates aged 19–70 years need to be sufficiently conscious and able to sign the treatment consent form and voluntary consent form. Pediatric patients aged 1–7 years could be recruited after signing an informed consent form by a legal surrogate (Guardian); pediatric patients aged 8–18 years need to be sufficiently conscious and voluntarily sign an informed consent form, and their legal surrogates (guardians) were also required to sign a written informed consent form.

Exclusion Criteria: An individual who meets any of the following criteria will be excluded from participation in this study: 1. Intracranial hypertension or disorder of consciousness; 2. Symptomatic heart failure or severe arrhythmia; 3. Symptoms of severe respiratory failure; 4. Complicated with other types of malignant tumors; 5. Diffuse intravascular coagulation; 6. Serum creatinine and/or blood urea nitrogen ≥1.5 times the normal value; 7. Suffering from septicemia or other uncontrollable infections; 8. Patients with uncontrollable diabetes; 9. Severe mental disorders; 10. Obvious and active intracranial lesions were detected by cranial magnetic resonance imaging (MRI); 11. Have received organ transplantation (excluding hematopoietic stem cell transplantation); 12. Reproductive-aged female patients with positive blood HCG test; 13. Screened to be positive for infection of hepatitis (including hepatitis B and C), AIDS, or syphilis; 14. Patients required to infuse autologous CAR T cells, with a tumor load higher than 30%.

We produced CAR T cells in the GMP laboratory of Beijing Gobroad Boren Hospital under strict GMP and SOP. In addition, the quality control of the CAR lentivirus was meticulously conducted at the same institution. Microbiological detections for endotoxin, bacteria, and mycoplasma were performed for each batch of lentivirus, along with a 7-day culture of the CAR vector for further microbiological examination. All operating procedures followed the principles of GMP and SOP in Beijing Gobroad Boren Hospital.

To be released, products must meet the following criteria: The final CAR T product should be free of bacterial and fungal growth, with no evidence of mycoplasma, or endotoxin below 10 pg/mL; the proportion of live cells should be greater than 70% by trypan blue exclusion; the proportion of CD3$^+$ T cells must exceed 60% by flow cytometry; the transfection rate of CAR to CD3$^+$ T cells must exceed 2% by flow cytometry. This criterion has been used in many clinical trials with different T cell qualities, and the CAR positivity rates of our CAR T products have been well above 2%.

Assessment of Disease Burden at Baseline and after CAR T Cell Infusion: To assess response and remission status, all patients underwent BM biopsy at baseline and on days 15 and 30 after CAR T cell infusion. Both morphological examinations of smears and flow cytometry were used to assess BM samples. Cerebrospinal fluid was evaluated by flow cytometry on day 30, and whenever necessary. CR or CRi, relapse, and MRD were defined following the National Comprehensive Cancer Network (NCCN) guidelines, version 2.2021. MRD$^-$ was defined as the absence of leukemia cells in BM determined by flow cytometry. The sensitivity of MRD analysis in our laboratory was 0.01% per NCCN guidelines.

The ASTCT consensus guidelines were used to grade CRS and ICANS[68]. Additionally, the EBMT consensus guidelines were used to grade GVHD[69]. For assessing other adverse events (AEs), including infections and hematologic toxicities, the CTCAE Version 5.0 was employed for appropriate grading. The management of CRS and neurotoxicity was carried out according to the established protocols specific to our center. Similarly, the management of GVHD, cytopenia, and infections followed the protocols developed and implemented within our center.

## Statistics & reproducibility

All of the data are representative of at least three independent experiments, if not stated otherwise. Data analysis and visualization were done using GraphPad Prism software. In the preclinical experiments of our study, no statistical method was used to predetermine sample size. No data were excluded from the analyses. In each animal experiment, we used mice of the same sex and age and randomized them into different experimental groups. Graphs show group means ± standard deviation (SD) of biological replicates or individual values unless otherwise noted. One-way ANOVA was used to compare more than two groups, and unpaired $t$-tests were used to compare two groups. We used the log-rank Mantel-Cox test to compare mouse survival curves. We considered a $P < 0.05$ as statistically significant.

In the clinical trial, the sample size was determined by clinical considerations and the 3 + 3 design. The infusion dose was based on the center's previous experience with CAR T cell therapy. Means with

corresponding standard deviations or medians with ranges were used for descriptive statistics, while safety and efficacy were analyzed using frequencies or percentages for categorical variables. Further information can be found in the supplementary information file.

### Reporting summary

Further information on research design is available in the Nature Portfolio Reporting Summary linked to this article.

## Data availability

Patient-related data not included in the manuscript may be restricted due to patient confidentiality. We have received IRB approval and have obtained consent to report data on these 4 patients. All data shared will be de-identified. The sequencing data have been deposited into the National Center for Biotechnology Information (NCBI) Sequence Read Archive (SRA) under accession code PRJNA1132480. All the other data are available within the article and its Supplementary Information. Source data are provided with this paper.

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

## Acknowledgements

We thank the patients who participated in this study and their families. We also thank all physicians, nurses, and other patient care providers involved in the study. We thank all members of our team for the critical discussion and suggestions. This study was supported by the National Key R&D Program of China (2021YFA1100703, grant to X.F.), CAMS Innovation Fund for Medical Sciences (CIFMS 2021-I2M-1-017, grant to X.F.), the National Natural Science Foundation of China (32170891, grant to X.F.; 82000191, grant to Q.N.), and Tianjin Municipal Science and Technology Commission Grant (21JCQNJC01750, grant to Q.N.). The sponsors of this study had no role in study design, data collection and analysis or manuscript writing.

## Author contributions

X.F. conceived the project. X.F. and G.W. directed the basic research and instructed the experiments. J.P. and T.W. directed the clinical study. S.Z., C.L., X.S., G.W., and X.F. designed the experiments. C.L., S.Z., X.S., and G.W. performed the experiments. Q.N. provided technical and experimental assistance. L.S. and K.T. provided experimental assistance. J.P., T.W., B.D., Z.L., J.X., J.D., Z.W., and Y.S. treated the patients. X.Y., Q.Z., X.X., and J.Z. contributed to the clinical lab diagnosis. J.P. and Z.T. participated in the design of the clinical trial. S.Z., G.W., and J.P. analyzed the clinical data. S.Z., C.L., X.S., G.W., Y.Z., Y.H., T.W., J.P., and X.F. prepared the manuscript. H.H., D.Z., S.F., and Z.H. provided help and critical suggestions.

## Competing interests

The authors declare no competing interests.

## Additional information

[1]State Key Laboratory of Experimental Hematology, National Clinical Research Center for Blood Diseases, Haihe Laboratory of Cell Ecosystem, Institute of Hematology & Blood Diseases Hospital, Chinese Academy of Medical Sciences & Peking Union Medical College, Tianjin, China. [2]Tianjin Institutes of Health Science, Tianjin, China. [3]Central laboratory, Fujian Medical University Union Hospital, Fuzhou, China. [4]Cytology Laboratory, Beijing GoBroad Boren Hospital, Beijing, China. [5]Department of Hematology, Beijing GoBroad Boren Hospital, Beijing, China. [6]Medical Laboratory, Beijing GoBroad Boren Hospital, Beijing, China. [7]Gobroad Research Center, Gobroad Medical Group, Beijing, China. [8]Department of Bone Marrow Transplantation, Beijing GoBroad Boren Hospital, Beijing, China. [9]Department of Pathogen Biology, School of Basic Medical Sciences, Tianjin Medical University, Tianjin, China. [10]Institute of Stem Cells, Health-Biotech (Tianjin) Stem Cell Research Institute Co., Ltd, Tianjin, China. [11]Department of Hematology, Peking University Shenzhen Hospital, Shenzhen Peking University-The Hong Kong University of Science and Technology Medical Center, Shenzhen, China. [12]State Key Laboratory of Experimental Hematology, Boren Clinical Translational Center, Department of Hematology, Beijing GoBroad Boren Hospital, Beijing, China. [13]These authors contributed equally: Shiyu Zuo, Chuo Li, Xiaolei Sun. ✉e-mail: wanggl2025@163.com; wut@gobroadhealthcare.com; panj@gobroadhealthcare.com; fengxiaoming@ihcams.ac.cn

