## [Peer Review File · Nature Communications]

C-JUN overexpression rescues impaired antigen signaling and enhances CAR-T efficacy in acute myeloid leukemiaREVIEWER COMMENTS

Reviewer #1 (Remarks to the Author); expert in CAR-T cells, preclinical and clinical AML:

Review of "Reviving ERK-CJUN circuitry rescues impaired antigen signaling and improves CART efficacy in AML" by Zuo et al.

In this paper, the authors combine in vitro preclinical experiments, in vivo xenograft studies and a first-in-human trial of anti-CD33 CART cells in 4 patients to arrive at their conclusion. The clinical trial is particularly impressive in showing that three of four patients achieved a remission, of whom two remained disease-free for over two years (after consolidative alloSCT).

The starting point for this work is the discrepancy between preclinical evidence of strong anti-AML activity of CART cells with the limited number of convincing clinical responses seen in the handful of clinical trials reported to date. The authors claim to have found that CART cells exposed to AML blasts show defective antigen receptor signaling compared with those exposed to ALL blasts, based on several cell line comparisons and (perhaps one?) primary AML/ALL comparison. CART-33 exposed to CD33-expressing ALL or AML cell lines then underwent bulk RNAseq which uncovered evidence of decreased activation including reduced ERK and NFAT signaling pathways. Largely due to the immune inhibitory effect of the poliovirus receptor molecule CD155 on AML cells, and that this could be reversed by overexpressing ZAP70 or C-JUN.

MAJOR COMMENTS:

1. The comparisons between ALL and AML were done using cell lines (using NALM6 or Raji as ALL, although notably Raji is not an ALL cell line; and using U937, THP1, and HL-60 as AML cell lines) that were engineered to express equivalent amount of CD33 (and in other experiments, CD123 or CLL1). The authors found in these experiments that CART-33 showed impaired activation upon exposure to the ALL cell lines than to the AML cell lines (while T cell numbers and leukemia cell apoptosis were unchanged). Most importantly, they then tested killing of primary AML and ALL by using CART-38 that can recognize the CD38 molecule on both histologies. How was cytotoxicity assessed (esp in the primary patient material, e.g. Fig 1F)? How many different primary AML/ALL experiments were done and how many are shown? The main issue here is that cell lines are not primary cells; and that the most robust conclusions will be generated from an appropriately sized comparison of primary AML with primary ALL. This cannot be done from a single AML donor and a single ALL donor.
2. RNAseq revealed decreased activation, ERK and NFAT pathways in CART33 exposed to AML cell line vs ALL cell line. There are many differences between the cell lines (which were really only matched for CD33 expression) e.g. integrins, co-stimulatory molecules etc therefore this analysis is flawed (but I concede it would be difficult to do it differently that the authors have attempted here). However, it is crucial to know how long the cell lines were exposed to CART cells?
3. Fig 2 shows acute events such as calcium flux and signaling molecule phosphorylation in AML-exposed CART cells after they had been sorted to 99% purity. How is this possible? Calcium flux is an acute and rapid event. Same comment for phosphoflow. Why did the authors not just do multi-parameter flow on the co-culture condition and gate on CART cells as appropriate?
4. Fig 2 RNAseq. How was the statistical comparison done? We are told in the figure legend that "each RNA sample was pooled from three experimental replicates". How many replicates were tested in total to arrive at the statistical differences? Stating "pooled" implies there was only one AML condition and one ALL condition. And if there were three experimental replicates (i.e. a total of six) then were these technical replicates or different T cell donors?
5. Novelty: CD155 has been previously reported to be associated with poor prognosis in AML, and the CD155:TIGIT interaction has already been described in AML immunotherapy.
6. The observation that PCR (CD155) is overexpressed in AML compared with ALL is based on RNAseq (Fig 3A) and flow cytometry (Fig 3B) of cell lines and primary samples. The most important data come from the primary samples. Why did the authors select just two primary samples per histology for RNAseq and three for flow? How did they decide which samples to use? This relates to the generalizability of their findings.
7. Rationale: the data up to and including Fig 3 would seem to support the production and testing of TIGIT KO (or some variant thereof) CART-33, rather than overexpressing LCK, ZAP70, CJUN or

FOS, as is described on p.10 line 12 and onwards. The rationale for this is therefore unclear.

8. Based on others' publications, it is really not surprising that overexpressing some of these transcription factors, signaling molecules, or cytokines would enhance CART cell function (Fig 4 and Fig 5). While this has not been shown in AML before (to my knowledge) this diminishes the novelty of this section since it has been shown in other tumor models previously.

9. Fig. 4: what do the recipients of C-JUN overexpressing CART cells die of? In several experiments the disease appears to be controlled until day 30 or so, but the animals die thereafter. Why? Panel 4F shows progressive expansion of CART cells without contraction, as would be expected if the leukemia were eradicated. Does this tell us that the C-JUN overexpressing CART cells are toxic?

10. Please show photomicrographs of patient marrows at pre- and post CART cell timepoints to illustrate cellularity.

MINOR COMMENTS:

1. Fig 1 legend: what does "PCDH" T cells mean?

2. What is meant by "assays were performed on day 10 after T-cell initial activation"? This sentence is in the legend of multiple figures.

3. Suppl Fig 2C is said to show total unphosphorylated ERK and C-JUN protein levels (both in the main text and in the figure legend) which are said to be comparable between CART cells exposed to ALL or AML, yet Supp Fig 2C shows higher C-JUN in AML-exposed to ALL-exposed CART cells. Please explain?

4. Fig 2C: since the focus of the paper is on AML, it is perhaps counter-intuitive to show changes as NALM6/U937 (where, I think, the reader has to invert the results). Perhaps this could be improved by specifying that down means down in AML, etc...

5. Fig 2C fourth panel from left, the scale appears to range from log₂ 10.8 to log₂ 12.90. What does this mean??

6. Supp 3A: How was ROS and NO quantified (what probes)?

7. The authors use "PVR" but it would be preferable to use the CD nomenclature (CD155).

8. Supp Fig 5B – how was C-JUN (y axis) detected? The construct does not appear to have a reporter. Was this direct intracellular flow staining for C-JUN? This point is particularly important in the patient products.

9. Consider moving the section that describes LD chemo, dose and sources of CART cells before the section on toxicity.

10. Patient 3 relapsed 1 year after CART cells followed by SCT and is reported to return to "disease-free status after undergoing SCT". This would be the patient's third HCT, by my count. Any treatment to get them into remission after the relapse post second HCT?

11. The second dose for patient 1 was the same number of CART cells as the first dose (0.5e6/kg)?

12. Table 1 states CART cell dose as 0.5 x 10⁵/kg but the text states 0.5 x 10⁶/kg.

13. Table 1 legend states "CNSL, central nervous system leukemia". But I cannot find other reference to CNS leukemia in this paper.

Reviewer #2 (Remarks to the Author); expert in PVR and AML therapy:

The authors describe that CAR T cells are less efficacious against myeloid AML cells than against lymphoblastic leukemia. Especially CAR T cells directed against CD33 on AML cells show less proliferation and activity and their CD19 counterpart against lymphoblastic leukemia cells. The authors investigate this difference in depth by analyzing various intracellular pathways, cytokine production and exhaustion markers. The major difference seems to be that the c-jun pathway is less active in CAR T cells against myeloid targets. Overexpression of c-jun improves CD33 CAR T cells function by activation of the ERK pathway.

These observations are certainly new and important for the development of next generation CAR T cells.

Comments:

1) RNA Seq analysis of CAR T cells directed against U937CD33 showed reduced levels of several cytokines such as IL-2, IL-21, LL-23A IFN γ (page 1, line 140). But later the authors tried to restore

CAR T cell function unsuccessfully only with incubation with IL-15 (page 10). Could addition of a more potent cocktail of cytokines increase CAR T cell function??

2) The authors blame expression of the immunosuppressive protein PVR on leukemic cells to be partly responsible for the reduced activity of CAR T cells. But as mentioned in the discussion this pathway is much more complex. On leukemic cells not only PVR but also PVRL2 is expressed. On the T-Cell side several positive and negative receptors such as TIGIT, PVRIG, DNAM are present. The authors should give more details on the relative importance of the receptors on CAR T cell activity and c-JUN and ERK expression and phosphorylation.

3) The authors present preliminary data on a Phase 1 clinical study with c-Jun overexpression CAR T cells. Four patients had been treated. Two patients obtained an MRD- CRI. From these early results it cannot be concluded that c-Jun CAR T cells are more clinically efficacious than regular CD33 CAR T cells. Therefore, I would suggest to omit the clinical part of the manuscript and mention that a clinical study is underway.

Reviewer #3 (Remarks to the Author); expert in immunology and signalling:

The manuscript by Zuo et al aims to explore the reasons why CAR T cell therapy is less efficient for AML than for other hematological malignancies such as ALL. To this end, the authors employed a collection of CARs targeting mainly CD33, but also alternatively CD123 or CLL1. Using model cell lines overexpressing CD33, U937 for AML and Nalm6 for ALL, the authors showed that killing of AML cells and subsequent CAR T cell activation is reduced independently of the antigen both in vitro and in vivo. Using a collection of cell AML vs ALL cell lines, they identify B7-H3, VISTA, TIM3 and PVR as molecules specifically upregulated in AML. Seeking for novelty, they decided to concentrate in PVR as possible candidate for regulating CAR T cell activity. Indeed, ko of PVR renders U397 susceptible to be killed by CAR T cells. At this point, the manuscript suffers from a lack of rational and coherence. The authors connect PVR overexpression in AML cells to reduced ERK activation in the CAR T cells, however mechanistic insights of this link are fully missing. The role of PVR high expression is not further followed.

The authors decided to increase CAR T cell activity by introducing signaling molecules of the TCR signaling cascade (LCK, ZAP70), the transcription factor c-Jun and IL15 (to increase expansion) in the plasmid containing the CAR. Why these molecules were chosen is not explained.

Overexpression of the introduced molecules is not shown. Nevertheless, co-introducing c-jun in the expression plasmid with the CAR results in better tumor control in vivo using either anti-CD33 or anti-CD123 CARs and using either 41BB or CD28 co-stimulatory molecules. These results are in line with previous reports on the beneficial effect of expressing c-jun in CAR T cells questioning the novelty of the present study:

<https://pubmed.ncbi.nlm.nih.gov/31802004/>, <https://pubmed.ncbi.nlm.nih.gov/34971569/>
<https://pubmed.ncbi.nlm.nih.gov/37467316/>

However, and in contrast with previous reports, including c-jun did not improved the exhaustion state of the CAR T cells, but their expansion, proliferation and killing activity by increasing the expression of genes associated to the ERK cascade (positive feedback loop?).

Lastly, the authors enrolled in a clinical trial in Phase I with a reduce number of patients. My expertise is not appropriated to evaluate the results of this part.

Points to be addressed:

1. My major concern is the lack of mechanistic insights of how high expression of PVR in AML cells is connected to poor CAR T cell performance. In my opinion providing this mechanism will definitively increase the impact of the manuscript. I acknowledge that identifying the ligand of PVR might be difficult, but at least the authors should explore the proposed ligands that are indeed mentioned in the discussion (CD226, TIGIT or CD96) by using blocking antibodies or KO these genes in U397 AML cells.

2. A second major concern is the presentation of the data. In a vast majority of assays (e.g Fig1 C-F, Fig2E-I, Fig3C-F, Fig5G-I, Fig6 F-N), experimental replicates are shown and not independent experiments performed with different donors (even though the authors claim to have done them). Showing experimental replicates of a single experiments is not acceptable. I anticipate that some of the statistically significant differences observed might be lost when biological replicates with

different donors are plotted. In line with this, statistical analysis should be revised, for example student-t test should not be used when more than 2 groups are plotted. SD should be shown in case of biological replicates.

3. Figure 1; what are PCHT T cells? It is never explained nor in the figure legend neither in the main text. In panel I, the gating strategy should be shown or at least explained in the text. In some panels the number of biological replicates is not depicted.

4. The text in lines 116-117 have to be revised, the CAR T cells obtained from the mice bearing U397 tumors exhibit reduced expression levels of X,y, z but not the mice!

5. Figure 2; The experimental setting of panels A-C must be explained. The CAR T cells were exposed to U397CD33 or Nalm6Cd33 for how long?

6. Figure 2, the sentence 136-137 has to be explained. The authors indeed analyzed ERK, NFAT or JNK pathway

7. Figure 2D-I; The rationale behind the experiment setting should be explained. Based on the method sections, CAR T cells were co-incubated with U397CD33 or Nalm6Cd33 for 12 hours, then sorted, and then stimulated again with parental U397 for 15-30 min "as indicated" (although is not indicated anywhere). I do not understand why this two step protocol if the authors aimed to assay activation of CAR T cells, why not directly co-incubating the CAR T cells with U397CD33 or Nalm6Cd33 and evaluate the signaling capacity? What is the aim of the first incubation time of 12 hours? My lecture from these experiments is that the first 12 hours the cells are reprogramed by the different signals obtained by the CAR and co-stimulatory receptors and this imprint the responses that we see in the second incubation. If the this is the case, this have to be rephrased. In addition, cells that do not encounter the tumor cells a second time or even better U397 cells lacking CD33 expression (which the authors have generated) should be used as negative controls. Likewise, CAR T cells stimulated via the TCR (anti-CD3) should be used as positive control as well as stimulation bypassing the TCR (PMA + iono.)

8. Figure 3; the authors identify PVR as the molecule to follow that might negatively regulate CAR T cell activation. The authors did Lost-of-functon(LOF) experiments to support this hypothesis. Gain-of-function experiments (GOT) should be also performed by overexpressing PVR in Nalm6 cells. The expectation is a reduction of killing when compared to Nalm6 not overexpressing PVR.

9. Figure 3F, please see comment 7

10. Figure 4; as mention above, the expression of the introduced molecules (LCK, ZAP70, c-jun and IL15) should be shown (flow cytometry, WB...)

11. The rational of how the authors have chosen these molecules LCK, ZAP70, c-jun) is missing, especially for LCK and ZAP70

12. The figure legend of Figure 4 is confusing: " Each line represents one treatment group. n=2-5 mice per group. Representative data from at least three experiments performed with different donors" How are n=2 mice divided in three experiments? Where the in vivo experiments done with different donors?

13. Figure 5: In previous reports, overexpression of c-jun revigorated or prevented exhaustion in CAR T cells, the authors do not see that effect as clear and justified this by using different target cells. Could the authors elaborate this explanation? Did the authors look for TIGIT that was one of the most obvious markers found downregulated in the previous reports?

14. Figure 6; In panels E-H, please see my comment #7. Did overexpression of c-jun only changed the levels of phosphor-proteins or also the expression of the proteins themselves (ERK, JNK, c-jun) because this could explain the increase in phosphor proteins.

15. Figure 6; panels K-N. The experiments should include the CAR T cells not overexpressing C-jun, I predict that in this cells UO126 also abolished activation and therefore the experiments is saying nothing to the link between overexpression of c-jun and ERK.

16. Discussion, lane 346: "Our results showed attenuated ERK signaling in CAR T cells in the presence of myeloid leukemia cells". First, not only ERK, but the authors showed calcium and ZAP70. The authors should discuss these data in the light of functional exhaustion, 12 hours of exposure to AML cells imprint the cells to reduced activation. Is this only by the CAR? Is TCR signaling normal?

17. Have the authors evaluate how overexpression of cjun affects tonic CAR signaling?

Minor points:

1. the figures should be called in consecutive order in the manuscript
2. The abbreviation c-JUN OE is not introduced

3. Lane 561; Phosphor-specific should be phosphor-specific

Reviewer #4 (Remarks to the Author); expert in clinical trial design and statistics:

The authors evaluate the safety of their proposed C-JUN-overexpressing CAR T cells in a Phase I trial, I have some comments on the clinical trial.

Major comments:

1. Based on the NCT04835519 registration, this study updated from a 3+3 design to a BOIN12 design, and the new design was planned to start on September 1, 2021, but in the paper, the authors stated that the four patients enrolled underwent CAR T cell infusion between April 13, 2021, and July 28, 2021. Were those 4 patients' clinical trials following the old protocol or the updated one? Please explain.
2. How the decision to suspend this study was made, based on a predefined rule or other reason, the authors should provide the reason in the result.
3. Since this study terminated too early, the objective defined in the protocol can not be evaluated based on the reported data, and the conclusion of "demonstrating efficacy in the first-in-human clinical trial" should be cautiously made.

Point-by-point responses to feedback on Manuscript

Manuscript Number: NCOMMS-23-43280A-Z

REVIEWER COMMENTS

Reviewer #1 (Remarks to the Author); expert in CAR-T cells, preclinical and clinical AML:

Review of “Reviving ERK-CJUN circuitry rescues impaired antigen signaling and improves CART efficacy in AML” by Zuo et al.

In this paper, the authors combine in vitro preclinical experiments, in vivo xenograft studies and a first-in-human trial of anti-CD33 CART cells in 4 patients to arrive at their conclusion. The clinical trial is particularly impressive in showing that three of four patients achieved a remission, of whom two remained disease-free for over two years (after consolidative alloSCT).

The starting point for this work is the discrepancy between preclinical evidence of strong anti-AML activity of CART cells with the limited number of convincing clinical responses seen in the handful of clinical trials reported to date. The authors claim to have found that CART cells exposed to AML blasts show defective antigen receptor signaling compared with those exposed to ALL blasts, based on several cell line comparisons and (perhaps one?) primary AML/ALL comparison. CART-33 exposed to CD33-expressing ALL or AML cell lines then underwent bulk RNAseq which uncovered evidence of decreased activation including reduced ERK and NFAT signaling pathways. largely due to the immune inhibitory effect of the poliovirus receptor molecule CD155 on AML cells, and that this could be reversed by overexpressing ZAP70 or C-JUN.

Response:

Thank you for your comments. We agree that more primary AML/ALL sample comparisons are needed, and have conducted additional experiments with more primary samples during revision, to confirm our findings. We have also made modifications based on the reviewer’s

suggestions, as described below in our point-by-point response, to improve our study.

MAJOR COMMENTS:

1. The comparisons between ALL and AML were done using cell lines (using NALM6 or Raji as ALL, although notably Raji is not an ALL cell line; and using U937, THP1, and HL-60 as AML cell lines) that were engineered to express equivalent amount of CD33 (and in other experiments, CD123 or CLL1). The authors found in these experiments that CART-33 showed impaired activation upon exposure to the ALL cell lines than to the AML cell lines (while T cell numbers and leukemia cell apoptosis were unchanged). Most importantly, they then tested killing of primary AML and ALL by using CART-38 that can recognize the CD38 molecule on both histologies. How was cytotoxicity assessed (esp in the primary patient material, e.g. Fig 1F)? How many different primary AML/ALL experiments were done and how many are shown? The main issue here is that cell lines are not primary cells; and that the most robust conclusions will be generated from an appropriately sized comparison of primary AML with primary ALL. This cannot be done from a single AML donor and a single ALL donor.

Response:

We thank the reviewer for this important point. We first identified tumor cells using the CD45/SSC gate and then used AML/ALL-specific markers (e.g. CD34, CD117, CD33, CD38, CD19, CD22) to distinguish blast cells from normal cells. We then used fluorescence-activated cell sorting (FACS) to sort primary AML and ALL samples expressing the same CD38 antigen. In the *in vitro* killing assays, the remaining primary AML and ALL cells were determined by the CD3⁻CD38⁺ circle gate. The results showed that CD38 CAR T cells were less effective in killing primary AML samples compared to ALL samples.

The primary AML/ALL experiments were performed multiple times with samples from different donors. We have provided new representative data and reanalyzed the statistics based on data from four to five different donors per group in Figure 1F.

We have newly added the gating strategy for assessing cytotoxicity in the revised Figure S1C, and have added the related description in lines 559–563, pages 25–26, in the Methods section of the revised manuscript.

2. RNAseq revealed decreased activation, ERK and NFAT pathways in CART33 exposed to AML cell line vs ALL cell line. There are many differences between the cell lines (which were really only matched for CD33 expression) e.g. integrins, co-stimulatory molecules etc therefore this analysis is flawed (but I concede it would be difficult to do it differently that the authors have attempted here). However, it is crucial to know how long the cell lines were exposed to CAR T cells?

Response:

We fully agree with the reviewer that there are many differences between cell lines, all of which may have different effects on CAR-T function and related pathways. We have tried to narrow down the candidate molecules using several strategies. First, to comprehensively and unbiasedly analyze which molecules in AML cells may mediate the inhibitory effects on CAR-T cells, we performed transcriptome sequencing; second, we paid more attention to the candidate genes that are consistently altered in the majority of AML cell lines studied, when compared to all the ALL lines. Third, the primary AML/ALL samples were also used to support the findings from cell lines. Finally, we then performed protein quantification and cell function rescue assays to verify the role of specific molecules in CAR T cell dysfunction in AML. Thus, we have progressively elucidated the mechanism of CAR T-cell defects in AML, despite the complexity that different tumor cell lines can cause. In light of the reviewer's comment, we accept that there may be some other differentially expressed molecules that also contribute to the suppressive mechanism, but have not yet been discovered in the present study, and we have mentioned this in the Discussion section of the revised manuscript.

CAR T cells were co-incubated with tumor cells for 2 days, and then we sorted CD8⁺ CAR⁺ T cells to perform RNA-seq.

We have added the related description in lines 117–119, page 7, in the Results section, lines 350–352, pages 16–17, in the Discussion section, and lines 609–611, page 27, in the Methods section of the revised manuscript.

3. Fig 2 shows acute events such as calcium flux and signaling molecule phosphorylation in

AML-exposed CART cells after they had been sorted to 99% purity. How is this possible? Calcium flux is an acute and rapid event. Same comment for phosphoflow. Why did the authors not just do multi-parameter flow on the co-culture condition and gate on CAR-T cells as appropriate?

Response:

We apologize for not clarifying this issue. We measured the acute events with a short (15-30 minutes) restimulation in CAR T cells pre-incubated with tumors for 12 hours for the following reasons.

First, we agree with the reviewer that calcium influx and phosphorylation were rapid events. In our preliminary experimental studies of others¹⁻³, phosphorylation of signaling proteins had rapid kinetics with a peak at about 15-30 min and then decayed. Therefore, the appropriate time point to measure a strong phosphorylation signal is several minutes after stimulation. However, direct measurement of calcium influx and protein phosphorylation after a short time of co-incubation (such as 15-30 minutes) is not appropriate in the present study, because the preliminary experiments have shown no difference in phosphorylation of ERK in CAR T co-incubated with U937 and Nalm6 cells for 15-30 minutes, suggesting that this is not an immediate defect in CAR T signaling upon AML exposure. Rather, it may take many hours for AML to induce extensive transcriptome and protein changes in CAR T cells, allowing CAR T cells to respond differentially to a second stimulation. This notion was supported by the fact that the phenotypic, functional, and transcriptomic changes (including the increased expression of ERK upstream regulators; Figures 1 and 2, Results section 1 and section 2) in CAR T cells became apparent only after many hours of incubation with leukemia cells.

Second, sorting by flow cytometry may allow the CAR T cells some time away from the leukemia cells so that they can be more sensitive to the second stimulation with regard to phosphorylation events.

We have clarified these issues and added the related data in the revised Supplementary Fig. 2C, and we have also added the related description in lines 138–141, page 8, in the Results section, and lines 521–527, page 24, in the Methods section of the revised manuscript.

4. Fig 2 RNAseq. How was the statistical comparison done? We are told in the figure legend that “each RNA sample was pooled from three experimental replicates”. How many replicates were tested in total to arrive at the statistical differences? Stating “pooled” implies there was only one AML condition and one ALL condition. And if there were three experimental replicates (i.e. a total of six) then were these technical replicates or different T cell donors?

Response:

We have now repeated the experiment with T cells from a different donor, and we now present data from two different donors, and the samples from each donor were pooled from three technical replicates.

We performed differentially expressed gene analysis using the DeSeq2 R package for two conditions⁴. We conducted GSEA enrichment analysis with the GSEA software (Broad Institute) as described⁵. For the GSEA of RNAseq data, false discovery rate (FDR) and normalized enrichment score (NES) were computed with the GSEA software.

We have added the related description in lines 609–618, pages 27–28, in the Methods section, and lines 721–723, pages 34–35, in the Figure 2 legend of the revised manuscript.

5. Novelty: CD155 has been previously reported to be associated with poor prognosis in AML, and the CD155:TIGIT interaction has already been described in AML immunotherapy.

Response:

We agree with the reviewer that CD155 has been reported to be highly expressed in myeloid leukemia cells, and associated with poor prognosis in AML⁶. However, the CAR T cell response may differ from the response of endogenous T cells, and the CD155 on CAR T cell response has not been previously reported.

To our knowledge, this study is the first to show that CD155 is expressed at higher levels in AML than in ALL, and that this difference is partly responsible for the lower anti-tumor function against AML than against ALL.

We have performed the blockade or knockout of CD226, TIGIT, or CD96. However, our preliminary results from antibody blockade or knockout of these receptors did not show a

significant effect on CAR T function. In addition, simultaneous blockade of both TIGIT and CD96 did not affect the cytotoxicity of CD33 CAR T cells. Thus, we believe that a novel finding is that the inhibitory effect of CD155 was independent of TIGIT/CD96 in CD33 CAR T cells. It is plausible that CD155 may interact with an as yet unidentified receptor on T cells to exert its suppressive effect.

We have added the data in the revised Figure 3H-L and Supplementary Figure 4C, and we have added the related description in lines 176–182, page 9, in the Results section, and lines 341–350, page 16, in the Discussion section of the revised manuscript.

6. The observation that PVR (CD155) is overexpressed in AML compared with ALL is based on RNAseq (Fig 3A) and flow cytometry (Fig 3B) of cell lines and primary samples. The most important data come from the primary samples. Why did the authors select just two primary samples per histology for RNAseq and three for flow? How did they decide which samples to use? This relates to the generalizability of their findings.

Response:

The decision to use only two primary samples for RNAseq was primarily due to the limited availability of primary samples in our study. We have now added two new patients in each group for RNAseq. The results also support our previous observations that CD155 is more highly expressed in AML patients compared to ALL patients.

We have updated the data in the revised Figure 3 A, and we have added the related description in lines 735–736, page 37, in the Figure 3A legend of the revised manuscript.

7. Rationale: the data up to and including Fig 3 would seem to support the production and testing of TIGIT KO (or some variant thereof) CART-33, rather than overexpressing LCK, ZAP70, CJUN or FOS, as is described on p.10 line 12 and onwards. The rationale for this is therefore unclear.

Response:

We appreciate your question regarding the rationale of our study. We have previously

performed the TIGIT blocking experiments, but the results showed no effect of this blocking on CAR T cell function. Because these results were preliminary, they were not included in the initial submission.

As suggested by the reviewer, we performed additional experiments to investigate the effects of TIGIT on CAR T cells. The results showed that neither TIGIT blockade nor knockout enhanced the cytotoxicity of CD33 CAR T cells against AML cells. Blockade of another CD155 inhibitory receptor CD96 also showed no effect. This suggests that the manipulation of TIGIT or CD96 was unlikely to promote CAR T efficacy in AML. Since we have not yet identified other inhibitory receptors of CD155 on CAR T cells, it is not feasible to target CAR T cells by knockout or RNA interference. Therefore, we believe that targeted engineering of the impaired antigen downstream signaling may be a more direct and simpler approach. As a result, overexpression of ZAP70 and C-JUN restored CAR-T function in AML, and C-JUN was found to induce long-term anti-tumor effects and prolong the survival of the mice.

We have updated the data in the revised Figure 3J–L, and we have added the related description in lines 185–190, page 10, in the Results section, and lines 347–350, page 16, in the Discussion section of the revised manuscript.

8. Based on others' publications, it is really not surprising that overexpressing some of these transcription factors, signaling molecules, or cytokines would enhance CART cell function (Fig 4 and Fig 5). While this has not been shown in AML before (to my knowledge) this diminishes the novelty of this section since it has been shown in other tumor models previously.

Response:

We acknowledge that most of the molecules in Figures 4 and 5 have been reported to enhance CAR T cell function. We hope the reviewer would agree that due to tumor specificity, the mechanism of resistance may be different for each tumor (such as AML), and there is an important need to identify specific resistance mechanisms and determine which molecules best enhance CAR T activity. Although ZAP70, LCK, C-JUN, and IL-15 have all been found to enhance anti-tumor effects when overexpressed⁷⁻¹¹, only C-JUN induced long-term anti-AML effects and prolonged the survival time of the mice in this study, and these results have not been

reported previously. Therefore we believe that one of the novel findings of the study is the mechanism-based identification of a factor that has superior performance and is most effective in reversing the resistance in AML at least among the bunch of candidates that have been studied.

The C-JUN activity in the previous study is more about exhaustion resistance. However, in the present study, the main mechanism of C-JUN is through a combination of reactivation of ERK and the transcriptional activity itself, thereby increasing a bunch of costimulatory molecules and cytokines, some of which have not been reported to be C-JUN targets. Therefore, to some degree, we have revealed some new mechanisms of C-JUN in promoting CAR T function in AML.

We have also added the related description in lines 353–356, 368–370, page 17, in the Discussion section of the revised manuscript.

9. Fig. 4: what do the recipients of C-JUN overexpressing CART cells die of? In several experiments the disease appears to be controlled until day 30 or so, but the animals die thereafter. Why? Panel 4F shows progressive expansion of CART cells without contraction, as would be expected if the leukemia were eradicated. Does this tell us that the C-JUN overexpressing CART cells are toxic?

Response:

We thank the reviewer for raising this issue. First, despite disease control at day 30 as demonstrated by blood sampling and tumor imaging, residual tumor cells may remain in bone marrow and other organs that were not subjected to flow cytometry analysis at regular time points. After day 30, the number of CAR T cells in the mice began to decline (Supplementary Figure 6I), leading to tumor progression and eventual death in some mice (Figure 4). In addition, Figure 4 shows that despite the advantages of C-JUN overexpression, there are limitations in the persistence of CAR T cells in AML.

To evaluate the potential toxicity of C-JUN CAR T cells, cytokine secretion was quantified in tail blood samples collected on days 7, 14, and 21 post-infusion. The results showed that C-JUN CAR T cells did not increase the cytokine levels. To accurately determine the cause of leukemia-related deaths, histological examination by hematoxylin and eosin staining was

performed on mice treated with control and C-JUN CAR T cells at their terminal stage. The pathological evaluation of vital organs, including the heart, liver, spleen, lung, and kidney, revealed no significant inflammatory cell infiltration, providing further evidence that C-JUN CAR T cells enhance therapeutic efficacy without increasing inflammatory organ toxicity.

The death of mice was also not caused by GVHD, because most of them did not show extensive expansion of CAR or non-CAR T cells in the blood, and the rarely occurred GVHD mice, characterized by reduced body weight, severely arched backs, reduced mobility, and disorganized hair curls, would have been excluded from the experiments.

We have added these new data in the revised Figure 4P, Q, and Supplementary Figure 6I, and we have also added the related description in lines 207–212, pages 10–11, in the Results section of the revised manuscript.

10. Please show photomicrographs of patient marrows at pre- and post CART cell timepoints to illustrate cellularity.

Response:

We have added the data in the revised Supplementary Figure 9A of the revised manuscript.

MINOR COMMENTS:

1. Fig 1 legend: what does “PCDH” T cells means?

Response:

“PCDH” T cells refer to T cells transduced with an empty pCDH vector used as a negative control. We have added the description in lines 445–446, page 21, in the Methods section of the revised manuscript.

2. What is meant by “assays were performed on day 10 after T-cell initial activation”? This sentence is in the legend of multiple figures.

Response:

This sentence refers to the fact that all *in vitro* and *in vivo* assays of CAR T cells were performed on day 10 after initial T cell activation with CD3/CD28 T cell activator, which is based on our previous experiments as well as the experience of others¹². We have included this sentence in lines 483–485, page 22, in the Methods section of the revised manuscript.

3. Suppl Fig 2C is said to show total unphosphorylated ERK and C-JUN protein levels (both in the main text and in the figure legend) which are said to be comparable between CART cells exposed to ALL or AML, yet Supp Fig 2C shows higher C-JUN in AML-exposed to ALL-exposed CART cells. Please explain?

Response:

We have corrected this error, and we also added the expression level of total ZAP70 protein in U937^{CD33} co-incubated and Nalm6^{CD33} co-incubated CAR T cells. We have changed this sentence to “Total ERK and C-JUN proteins in U937^{CD33} co-incubated CAR T cells are comparable to or higher than those in Nalm6^{CD33} co-incubated CAR T cells, while total ZAP70 protein was slightly lower”.

We have added the new data in the revised Supplementary Figure 2G and 2H, and we have included this sentence in lines 152–155, page 8, in the Results section of the revised manuscript.

4. Fig 2C: since the focus of the paper is on AML, it is perhaps counter-intuitive to show changes as NALM6/U937 (where, I think, the reader has to invert the results). Perhaps this could be improved by specifying that down means down in AML, etc...

Response:

We have specified “down” or “up” as “down in AML” or “up in AML” in the revised Figure 2B and Figure 3A.

5. Fig 2C fourth panel from left, the scale appears to range from log2 10.8 to log2 12.90. What does this mean??

Response:

Thank you for the question. We have now repeated the experiment with T cells from a different donor, and we now present data from two different donors, and the samples from each donor were pooled from three technical replicates. Differentially expressed genes were identified using the DeSeq2 R package for two conditions. The genes now shown in the heat maps meet the parameters: fold change ≥ 1.5 -fold in each of the two biological replicates. The heat maps are based on \log_2 -transformed expression values. Row z-scores are shown and were calculated using TBtools.

We have added the new data in the revised Figure 2A and 2B, and we have added the related description in lines 612–618, pages 27–28, in the Methods section, and lines 721–723, pages 34–35, in the Figure 2 legend of the revised manuscript.

6. Supp 3A: How was ROS and NO quantified (what probes)?

Response:

We measured ROS levels using a ROS measurement kit (Beyotime). Cells were plated and loaded with 10 $\mu\text{mol/L}$ DCFH-DA at 37 °C for 30 minutes in the dark and then washed three times with serum-free medium. Intracellular NO levels were measured using DAF-FM DA (Beyotime). Cells were loaded with 5 $\mu\text{mol/L}$ DAF-FM DA at 37 °C for 30 minutes in the dark. The cells were washed and analyzed by flow cytometry.

We have added the related description in lines 544–546, page 25, in the Methods section of the revised manuscript.

7. The authors use “PVR” but it would be preferable to use the CD nomenclature (CD155).

Response:

We have changed “PVR” to “CD155” throughout the manuscript.

8. Supp Fig 5B – how was C-JUN (y axis) detected? The construct does not appear to have a reporter. Was this direct intracellular flow staining for C-JUN? This point is particularly

important in the patient products.

Response:

We have verified C-JUN expression preclinically by sequencing and by intracellular flow staining with C-JUN monoclonal antibody (Clone 60A8; Cell Signaling Technology). In addition, we have also confirmed the successful expression of C-JUN by intracellular flow staining in available CAR T products from patients. We will also consider measuring C-JUN expression in all patients and at different time points in further studies.

We have added the description in line 504–507, page 23, in the Methods section of the revised manuscript.

9. Consider moving the section that describes LD chemo, dose and sources of CART cells before the section on toxicity.

Response:

We have moved the section describing LD chemo, dose, and sources of CAR T cells before the section on toxicity in lines 284–286, page 14, in the Results section of the revised manuscript.

10. Patient 3 relapsed 1 year after CART cells followed by SCT and is reported to return to “disease-free status after undergoing SCT”. This would be the patient’s third HCT, by my count. Any treatment to get them into remission after the relapse post second HCT?

Response:

Patient 3 relapsed after one year of CAR T-cell therapy, the patient was subsequently treated with Selinexor and Artesunate, and after remission, the patient underwent stem cell transplantation. We have added the related description in lines 314–316, page 15, in the Results section of the revised manuscript.

11. The second dose for patient 1 was the same number of CART cells as the first dose

(0.5e6/kg)?

Response:

The second dose of CAR T cells for patient 1 was not the same as the first dose, and the second infusion dose was $3.76 \times 10^6/\text{kg}$. We have added this in line 848, page 48, in the Table 1 legend of the revised manuscript.

12. Table 1 states CART cell dose as $0.5 \times 10^5/\text{kg}$ but the text states $0.5 \times 10^6/\text{kg}$.

Response:

We have now corrected the infusion dose from $0.5 \times 10^5/\text{kg}$ to $0.5 \times 10^6/\text{kg}$ in Table 1 of the revised manuscript.

13. Table 1 legend states “CNSL, central nervous system leukemia”. But I cannot find other reference to CNS leukemia in this paper.

Response:

We have removed this in the Table 1 legend of the revised manuscript.

Reviewer #2 (Remarks to the Author); expert in PVR and AML therapy:

The authors describe that CAR T cells are less efficacious against myeloid AML cells than against lymphoblastic leukemia. Especially CAR T cells directed against CD33 on AML cells show less proliferation and activity and their CD19 counterpart against lymphoblastic leukemia cells. The authors investigate this difference in depth by analyzing various intracellular pathways, cytokine production and exhaustion markers. The major difference seems to be that the c-jun pathway is less active in CAR T cells against myeloid targets. Overexpression of c-jun improves CD33 CAR T cells function by activation of the ERK pathway.

These observations are certainly new and important for the development of next generation

CAR T cells.

Response:

We are grateful to the reviewer for the kind summary of our study. We also appreciate the reviewer's suggestions. We believe that incorporating their suggestions, as described below in our point-by-point response, has improved the manuscript.

Comments:

1) RNA Seq analysis of CAR T cells directed against U937CD33 showed reduced levels of several cytokines such as IL-2, IL-21, IL-23A, IFN γ (page 1, line 140). But later the authors tried to restore CAR T cell function unsuccessfully only with incubation with IL-15 (page 10). Could addition of a more potent cocktail of cytokines increase CAR T cell function??

Response:

We agree that adding a more potent cytokine cocktail could potentially enhance CAR T cell function. In our revised manuscript, we have performed a *in vivo* experiment using IFN- γ -overexpressing CD33 CAR T. However, our results showed that these modified CAR T cells did not effectively control tumor growth and unexpectedly resulted in a shorter survival than the control CD33 CAR T cells.

Overexpression of IL-15 has previously been shown to enhance CAR T cell expansion in various tumor types^{7,8}. In addition, overexpression of IL-15 alone in CLL-1 CAR T cells has been reported to promote CAR T expansion and partially control tumor progression in AML, but upregulates TNF- α secretion and causes severe lung and liver damage, and improves mice survival only after using a TNF- α inhibitor⁹. Overexpression of both IL-15 and IL-21 has been shown to further enhance the anti-tumor effect¹³. Thus, while we recognize the potential of a more potent cytokine cocktail to enhance CAR T cell function and reduce tumor burden, we are concerned about the possibility of inducing severe acute toxicity due to excessive cytokine production and inflammation.

Furthermore, in addition to the reduced multiple cytokines, U937^{CD33}-exposed CAR T cells also showed significant reductions in many cytotoxic molecules and co-stimulatory molecules.

Therefore, we suspect that overexpression of cytokines alone may have a limited effect. Based on these considerations, we did not use a potent cytokine cocktail in our study. However, we believe that exploring the delicate balance between therapeutic efficacy and potential toxicity remains a critical issue for future research.

We have updated the data in the revised Supplementary Figure 5B–D, and we have also added the related description in lines 191, 193–194, page 10, in the Results section, and lines 358–359, 363–365, page 17, in the Discussion section of the revised manuscript.

2) The authors blame expression of the immunosuppressive protein PVR on leukemic cells to be partly responsible for the reduced activity of CAR T cells. But as mentioned in the discussion this pathway is much more complex. On leukemic cells not only PVR but also PVRL2 is expressed. On the T-Cell side several positive and negative receptors such as TIGIT, PVRIG, DNAM are present. The authors should give more details on the relative importance of the receptors on CAR T cell activity and c-JUN and ERK expression and phosphorylation.

Response:

We agree that the CD155-TIGIT axis between leukemic cells and CAR T cells is complex. Based on the reviewer's suggestions, we have further investigated this issue.

(1) We have found that CD155 (PVR), but not CD112 (PVRL2), inhibited the effector function of CD33 CAR T cells in AML. Upon blocking CD155 expression, we observed increased anti-tumor efficacy of CAR T cells. In contrast, revision experiments suggest that blocking CD112 did not affect the anti-tumor activity of CAR T cells.

(2) We also investigated the interactions between CD155 and its receptors TIGIT, CD96, and CD226. However, our results from blockade or knockout of these receptors did not show a significant effect on CAR T function. In addition, simultaneous blockade of both TIGIT and CD96 did not affect the cytotoxicity of CD33 CAR T cells. Thus, we suggest that the inhibitory effect of CD155 was independent of TIGIT/CD96 in CD33 CAR T cells. It is plausible that CD155 may interact with an as yet unidentified receptor on T cells to exert its suppressive effect, which requires further investigation

(3) Finally, in the initial submission, we performed phospho-flow assays to analyze ERK

signaling in CAR T cells. Interestingly, CD155 reduced the phosphorylation of ERK in CAR T cells, indicating a suppressive effect on their effector function. During the revision, we performed additional experiments and found that CD155 reduced the phosphorylation of C-JUN in CAR T cells, but not ZAP70. In addition, we found that CD155 did not affect the total ERK, C-JUN, or ZAP70 protein expression.

We have added the related data in the revised Figure 3F–L and Supplementary Figure 4A–C. We have added the related description in lines 172–174, 176–182, page 9, in the Results section, and lines 341–350, page 16, in the Discussion section of the revised manuscript,

3) The authors present preliminary data on a Phase 1 clinical study with c-Jun overexpression CAR T cells. Four patients had been treated. Two patients obtained an MRD- CRi. From these early results it cannot be concluded that c-Jun CAR T cells are more clinically efficacious than regular CD33 CAR T cells. Therefore, I would suggest to omit the clinical part of the manuscript and mention that a clinical study is underway.

Response:

We appreciate the reviewer's critique of our preliminary clinical data and the insightful suggestions. Regarding CD33 CAR-T treatment for AML, published data to date have shown limited expansion of CAR T cells and a lack of remission in the majority of treated patients^{14,15}. Therefore, there is an urgent need to develop an effective CAR-T therapy approach in the clinic, so we expect that these clinical data may be of great interest to the physician researchers in this field. In addition, the CAR-T activity in the human and mouse microenvironment would be different, so we think it is important information to observe the activity of CAR-T cells in patients.

We agree that given the small sample size and early stage of this Phase I study, it is premature to conclude that C-JUN CAR T cells are more effective than regular CD33 CAR T cells in AML. We also acknowledge the limitation of the single-arm design, which prevents a direct comparison between C-JUN and non-C-JUN CAR T cells.

We have added the related description in lines 409, 420–422, page 19, in the Discussion section of the revised manuscript.

Reviewer #3 (Remarks to the Author); expert in immunology and signalling:

The manuscript by Zuo et al aims to explore the reasons why CAR T cell therapy is less efficient for AML than for other hematological malignancies such as ALL. To this end, the authors employed a collection of CARs targeting mainly CD33, but also alternatively CD123 or CLL1. Using model cell lines overexpressing CD33, U937 for AML and Nalm6 for ALL, the authors showed that killing of AML cells and subsequent CAR T cell activation is reduced independently of the antigen both in vitro and in vivo. Using a collection of cell AML vs ALL cell lines, they identify B7-H3, VISTA, TIM3 and PVR as molecules specifically upregulated in AML. Seeking for novelty, they decided to concentrate in PVR as possible candidate for regulating CAR T cell activity. Indeed, ko of PVR renders U397 susceptible to be killed by CAR T cells. At this point, the manuscript suffers from a lack of rational and coherence. The authors connect PVR overexpression in AML cells to reduced ERK activation in the CAR T cells, however mechanistic insights of this link are fully missing. The role of PVR high expression is not further followed.

The authors decided to increase CAR T cell activity by introducing signaling molecules of the TCR signaling cascade (LCK, ZAP70), the transcription factor c-Jun and IL15 (to increase expansion) in the plasmid containing the CAR. Why these molecules were chosen is not explained. Overexpression of the introduced molecules is not shown. Nevertheless, co-introducing c-jun in the expression plasmid with the CAR results in better tumor control in vivo using either anti-CD33 or anti-CD123 CARs and using either 41BB or CD28 co-stimulatory molecules. These results are in line with previous reports on the beneficial effect of expressing c-jun in CAR T cells questioning the novelty of the present study:

<https://pubmed.ncbi.nlm.nih.gov/31802004/>, <https://pubmed.ncbi.nlm.nih.gov/34971569/>

<https://pubmed.ncbi.nlm.nih.gov/37467316/>

However, and in contrast with previous reports, including c-jun did not improved the exhaustion state of the CAR T cells, but their expansion, proliferation and killing activity by increasing the

expression of genes associated to the ERK cascade (positive feedback loop?).

Lastly, the authors enrolled in a clinical trial in Phase I with a reduce number of patients. My expertise is not appropriated to evaluate the results of this part.

Response:

We appreciate your question about the mechanistic insights of CD155 and the novelty of C-JUN overexpression in CAR T cells. In the revision, the CD155-related mechanisms have been further explored, and the specific new information on C-JUN overexpression provided by this study is further elaborated in the Discussion section. We have carefully considered and incorporated your comments into our work and believe that they add to the strength and clarity of our paper.

Points to be addressed:

1. My major concern is the lack of mechanistic insights of how high expression of PVR in AML cells is connected to poor CAR T cell performance. In my opinion providing this mechanism will definitively increase the impact of the manuscript. I acknowledge that identifying the ligand of PVR might be difficult, but at least the authors should explore the proposed ligands that are indeed mentioned in the discussion (CD226, TIGIT or CD96) by using blocking antibodies or KO these genes in U397 AML cells.

Response:

We appreciate the reviewer's insightful comments and admit it is important to further study the mechanism of CD155.

Our additional exploration showed blocking CD112 did not have a significant role in inhibiting the effector function of CD33 CAR T cells in AML.

We also follow the reviewer's suggestion to investigate the effect of blocking or knockout CD155 receptors (TIGIT, CD96, and CD226). However, our preliminary results from blockade or knockout of these receptors did not show a significant effect on CAR T function. In addition, simultaneous blockade of both TIGIT and CD96 did not affect the cytotoxicity of CD33 CAR T cells. Thus, we suggest that the inhibitory effect of CD155 was independent of TIGIT/CD96

in CD33 CAR T cells. It is plausible that CD155 may interact with an as yet unidentified receptor on T cells to exert its suppressive effect, which requires further investigation

We have added the related description in lines 176–182, page 9, in the Results section, and lines 341–350, page 16, in the Discussion section of the revised manuscript.

2. A second major concern is the presentation of the data. In a vast majority of assays (e.g Fig1 C-F, Fig2E-I, Fig3C-F, Fig5G-I, Fig6 F-N), experimental replicates are shown and not independent experiments performed with different donors (even though the authors claim to have done them). Showing experimental replicates of a single experiments is not acceptable. I anticipate that some of the statistically significant differences observed might be lost when biological replicates with different donors are plotted. In line with this, statistical analysis should be revised, for example student-t test should not be used when more than 2 groups are plotted. SD should be shown in case of biological replicates.

Response:

In response to your suggestions, we have made the following changes in the revised manuscript.

First, there are both technical and biological replicates in our study. We have replaced the technical replicates with biological replicates, as recommended in Figures 3, 5, and 6, and the number of biological replicates for each panel has been indicated in the figure legends. Second, regarding your concern about the method of statistical analysis, we have now changed to one-way ANOVA analysis for comparisons involving more than two groups. For comparisons between the two groups, we used the unpaired *t*-test. Finally, we have presented the results of biological replicates as mean \pm SD.

When we made more biological replicates, we found that IL-2 expression was higher in C-JUN CAR T cells than in control CAR T cells *in vitro*, so we changed the data in the revised Figure 5H.

We have added the description in lines 223–224, page 11, in the Results section, lines 670–672, page 30, in the Methods section, and the figure legend of the revised manuscript.

3. Figure 1; what are PCDH T cells? It is never explained nor in the figure legend neither in the main text. In panel I, the gating strategy should be shown or at least explained in the text. In some panels the number of biological replicates is not depicted.

Response:

“PCDH” T cells refer to T cells transduced with an empty pCDH vector used as a negative control. We have added the description in lines 445–446, page 21, in the Methods section of the revised manuscript.

We have added the gating strategy of panel I in the revised Supplementary Figure 1I, and we have also added the related description of the number of biological replicates in the Figure legend of the revised manuscript.

4. The text in lines 116-117 have to be revised, the CAR T cells obtained from the mice bearing U937 tumors exhibit reduced expression levels of X,y, z but not the mice!

Response:

We have changed the sentence to “After 3-5 days, CAR T cells in mice bearing U937^{CD33} tumors had fewer cell counts, higher apoptotic rates, and lower expression of CD25, granzyme B, IFN- γ , and IL-2 than in mice bearing Nalm6^{CD33} tumors” in lines 109–111, pages 6–7, in the Results section of the revised manuscript.

5. Figure 2; The experimental setting of panels A-C must be explained. The CAR T cells were exposed to U397CD33 or Nalm6Cd33 for how long?

Response:

In the revised panels A and B of Figure 2, CAR T cells were exposed to tumor cells for 2 days, and then we sorted CD8⁺ CAR⁺ T cells by FACS to perform bulk RNA-seq.

We have added the related description in lines 117–119, page 7, in the Results section, and lines 609–610, page 27, in the Methods section of the revised manuscript.

6. Figure 2, the sentence 136-137 has to be explained. The authors indeed analyzed ERK, NFAT or JNK pathway

Response:

In our manuscript, we stated that “T cell-specific ERK, NFAT, or JNK_MAPK pathways were not provided in the GSEA database and therefore could not be analyzed in our manuscript”. We initially mentioned this limitation as we aimed to identify T cell-specific pathways, but the lack of corresponding T cell-specific datasets in GSEA. After careful consideration, we acknowledge that this statement may have been unclear and have subsequently removed it from our revised manuscript.

7. Figure 2D-I; The rationale behind the experiment setting should be explained. Based on the method sections, CAR T cells were co-incubated with U397CD33 or Nalm6Cd33 for 12 hours, then sorted, and then stimulated again with parental U397 for 15-30 min “as indicated” (although is not indicated anywhere). I do not understand why this two step protocol if the authors aimed to assay activation of CAR T cells, why not directly co-incubating the CAR T cells with U397CD33 or Nalm6Cd33 and evaluate the signaling capacity? What is the aim of the first incubation time of 12 hours? My lecture from these experiments is that the first 12 hours the cells are reprogramed by the different signals obtained by the CAR and co-stimulatory receptors and this imprint the responses that we see in the second incubation. If the this is the case, this have to be rephrased. In addition, cells that do not encounter the tumor cells a second time or even better U397 cells lacking CD33 expression (which the authors have generated) should be used as negative controls. Likewise, CAR T cells stimulated via the TCR (anti-CD3) should be used as positive control as well as stimulation bypassing the TCR (PMA + iono.)

Response:

We apologize for not clarifying the experimental rationale. As you correctly surmised, the intent behind our two-step protocol is indeed to study the reprogramming events that occur within CAR T cells during the first 12 hours of co-incubation with myeloid and lymphoid tumor cells. This process allows the CAR T cells to encounter different signals generated by the CAR

and co-stimulatory receptors, which we believe will shape the subsequent responses seen in the second stimulation. Indeed, in our preliminary experiments, CAR T cells showed no difference in the phosphorylation of ERK after a short time of co-incubation (such as 15-30 min) with U937 or Nalm6 cells, so we suspect that this is not an immediate defect in CAR T signaling upon AML exposure. We thank the reviewer for this precise rephrasing and have included this rationale in the revised manuscript.

We thank the reviewer for this very helpful suggestion for setting up positive and negative controls. Following his suggestion, new experiments were performed. Our results showed that U937^{CD33} pre-incubated CAR T cells that received the second stimulation with PMA or anti-CD3 had lower ERK and C-JUN phosphorylation levels than Nalm6^{CD33} pre-incubated CAR T cells. Probably because PMA stimulation bypasses the TCR, the levels of ZAP70 phosphorylation in U937^{CD33} pre-incubated CAR T cells stimulated with PMA were comparable to the negative control of U937^{CD33-KO} stimulation. When stimulated with anti-CD3, ZAP70 phosphorylation levels in U937^{CD33} pre-incubated CAR T cells were significantly lower than in Nalm6^{CD33} pre-incubated CAR T cells. These results are consistent with what we observed when U937 cells were used for the second stimulation.

We have added these data in the revised Supplementary Figure 2D–F, and we have added the related descriptions in lines 138–145, 149–152, page 8, in the Results section, and lines 521–527, page 24, in the Methods section of the revised manuscript.

8. Figure 3; the authors identify PVR as the molecule to follow that might negatively regulate CAR T cell activation. The authors did Lost-of-function (LOF) experiments to support this hypothesis. Gain-of-function experiments (GOT) should be also performed by overexpressing PVR in Nalm6 cells. The expectation is a reduction of killing when compared to Nalm6 not overexpressing PVR.

Response:

As recommended by the reviewer, we have successfully overexpressed CD155 in Nalm6 cells by lentivirus and verified CD155 expression at the protein level by flow cytometry. We then co-cultured CD19 CAR T cells with control Nalm6 cells and CD155 overexpressing

(CD155 OE) Nalm6 cells. The results showed a notable decrease in the cytotoxicity of CD19 CAR T cells against CD155 OE Nalm6 cells as compared to control Nalm6 cells.

We have added the data in the revised Figure 3G, and we have also added the related description in lines 174–175, page 9, in the Results section of the revised manuscript.

9. Figure 3F, please see comment 7

Response:

Following the reviewer's suggestion, we performed the new experiments that included positive and negative controls. Our results showed that U937^{CD155 KO} pre-incubated CAR T cells stimulated with either anti-CD3 or PMA had higher ERK phosphorylation levels than U937^{CD155 WT} pre-incubated CAR T cells. We also examined C-JUN and ZAP70 phosphorylation when re-stimulated with these negative and positive controls and found that U937^{CD155 KO} pre-incubated CAR T cells had increased C-JUN phosphorylation but not ZAP70 phosphorylation.

We have added the data in the revised Figure 3F and Supplementary Figure 4A, and we have also added the related description in lines 172–173, page 9, in the Results section of the revised manuscript.

10. Figure 4; as mention above, the expression of the introduced molecules (LCK, ZAP70, c-jun and IL15) should be shown (flow cytometry, WB...)

Response:

We have shown the expression of the introduced molecules (ZAP70, LCK, C-JUN, and IL-15) in the revised Supplementary Figure 5A, and we have also added the description in lines 504–507, page 23, lines 576–580, page 26, in the Methods section of the revised manuscript.

11. The rational of how the authors have chosen these molecules LCK, ZAP70, c-jun) is missing, especially for LCK and ZAP70

Response:

We apologize for missing this very important information.

LCK, ZAP70, C-JUN, and C-Fos are well known to be key players in T cell activation and function. In particular, LCK and ZAP70 are T cell receptor proximal signaling essential for initiation and transduction of the downstream antigen signal. Upon TCR engagement, LCK phosphorylates the TCR and CD3, leading to recruitment and phosphorylation of ZAP70¹⁶. This leads to further downstream signaling events that ultimately result in T cell activation, proliferation, and effector functions. Notably, both C-JUN and C-Fos are components of the AP-1 transcription factor complex, which is a critical regulator of T cell activation and effector function¹⁷.

In our study, we found that CD33 CAR T cells exhibited impaired activation and decreased expression of effector molecules when exposed to AML cells. In addition, RNA-seq analysis revealed profound defects in TCR calcium and ERK signaling pathways. Phosphorylation assays also showed lower levels of ZAP70, ERK, and C-JUN. LCK is an upstream regulator of ZAP70. Therefore, we selected ZAP70, LCK, C-JUN, and C-Fos for functional enhancing experiments.

We have added the related descriptions in lines 186–190, page 10, in the Results section of the revised manuscript.

12. The figure legend of Figure 4 is confusing: “Each line represents one treatment group. n=2–5 mice per group. Representative data from at least three experiments performed with different donors” How are n=2 mice divided in three experiments? Where the *in vivo* experiments done with different donors?

Response:

We apologize for the confusion this has caused. To clarify, the *in vivo* experiments shown in Figure 4 are representative data obtained from independent experiments using T cells from two different donors. For each experiment, we used 2 to 5 mice per group. The statement “Each line represents one treatment group. n=2–5 mice per group. Representative data from at least three experiments performed with different donors” was intended to convey that the data shown in Figure 4 are a representation of consistent findings across multiple experiments.

We have now clarified the number of biologically independent samples in each experiment and also clarified that the data shown in Figure 4 are representative of two independent experiments. We have added the related description in lines 760–761, 771–774, page 39, in the Figure 4 legend of the revised manuscript.

13. Figure 5: In previous reports, overexpression of c-jun revigorated or prevented exhaustion in CAR T cells, the authors do not see that effect as clear and justified this by using different target cells. Could the authors elaborate this explanation? Did the authors look for TIGIT that was one of the most obvious markers found downregulated in the previous reports?

Response:

As mentioned by the reviewer, previous studies showed that overexpression of C-JUN can reinvigorate CAR T cells and resist exhaustion.

In our study, we did not observe the effect of C-JUN overexpression in resisting exhaustion, because we did not observe the decrease of inhibitory receptors PD-1, TIM-3, LAG-3, and the transcription factor TOX in C-JUN CAR T cells. We propose the following notions to explain this discrepancy. First, the study by Crystal L. Mackall et al. was designed to address the issue of T-cell exhaustion and incorporated antigenic targets or co-stimulatory molecules in CAR constructs that are prone to severe exhaustion, such as HA-28z and GD2-28z^{18,19}. In this context, overexpression of C-JUN was found to be effective in rescuing T-cell exhaustion. In contrast, our study used CD33 bbz, which was found to be less susceptible to exhaustion compared to CD33 28z CAR T cells²⁰. Second, the short persistence time of CD33 CAR T in AML-bearing mice may prevent the observation of exhaustion for a longer time. Third, the lack of exhaustion of CAR T in AML may be caused by the property of this tumor and the resistance mechanisms. As we show in our study, the CAR T in AML mainly manifested as hypoactivation and decreased effector program, indicating an insufficient stimulatory signal. This specific defect of CAR-T may lead to a different mode of action of C-JUN overexpression.

As suggested by the reviewer, we further investigated the expression of TIGIT. The results showed that C-JUN CAR T had slightly higher TIGIT expression in both CD4⁺ and CD8⁺ CAR T cells, which is consistent with the increased PD-1 expression that we presented previously.

We have added the data in the revised Figure 5F, and we have also added the related description in line 222, page 11, in the Results section, and lines 379–389, page 18, in the Discussion section of the revised manuscript.

14. Figure 6; In panels E-H, please see my comment #7. Did overexpression of c-jun only changed the levels of phosphor-proteins or also the expression of the proteins themselves (ERK, JNK, c-jun) because this could explain the increase in phosphor proteins.

Response:

As suggested by the reviewer, we performed re-stimulation with U937^{CD33KO} (negative control), PMA, or anti-CD3. The results showed that C-JUN CAR T cells re-stimulated with PMA or anti-CD3 or had higher levels of p-ERK, p-C-JUN, and p-JNK (only with anti-CD3) than control CAR T cells.

We have now performed *in vitro* experiments comparing the expression of the proteins themselves (ERK, JNK, C-JUN) in C-JUN CAR and control CAR T cells in the presence of myeloid leukemia cells. Our results showed that there were no differences in ERK and JNK expression levels between the two groups, whereas overexpression with C-JUN increased the expression of C-JUN protein. Therefore, we speculate that the increased phosphorylation of ERK and JNK may not be determined by their protein expression, but rather may be influenced by the increased rate of phosphorylation induced by upstream activating signals. The benefits of C-JUN overexpression may be mediated by both an increased expression of the C-JUN protein itself and an increased C-JUN phosphorylation.

We have added the data in the revised Supplementary Figure 7D–E, and we have also added the description in lines 251–258, page 12, in the Results section of the revised manuscript.

15. Figure 6; panels K-N. The experiments should include the CAR T cells not overexpressing C-jun, I predict that in this cells U0126 also abolished activation and therefore the experiments is saying nothing to the link between overexpression of c-jun and ERK.

Response:

We have additionally performed experiments with a high concentration of 50 μ M U0126 treatment of control CD33 CAR T cells and found that U0126 also suppresses activation, as confirmed by reduced expression levels of 4-1BB, IL-2, IL-21, and IFN- γ .

To understand the outcome of the C-JUN-induced increase of ERK activation, we co-incubated C-JUN CAR T cells with U937 cells and added different concentrations of U0126 inhibitor (1 μ M, 5 μ M, and 10 μ M), and found that at a concentration of 5 μ M U0126 inhibitor, C-JUN CAR T cells exhibited comparable levels of ERK phosphorylation as control CAR T cells without U0126 treatment. This specific situation allows a more direct assessment of the contribution of C-JUN-induced increase in ERK activation to T cell activation in this condition, U0126 only suppressed 4-1BB, IL-2, and IFN- γ but not CD28, CD86, and IL-21 protein expression in C-JUN CAR T cells, suggesting that 4-1BB, IL-2, and IFN- γ are dependent on the increased C-JUN-mediated ERK activity. In contrast, CD28, CD86, and IL-21 protein is unlikely dependent on the increase in ERK activity, but rather on direct C-JUN-mediated transcriptional activation.

We have added the data in the revised Figure 6M–O, and Supplementary Figure 8A–B, and we have also added the related descriptions in lines 265–277, page 13, in the Results section of the revised manuscript.

16. Discussion, line 346: “Our results showed attenuated ERK signaling in CAR T cells in the presence of myeloid leukemia cells”. First, not only ERK, but the authors showed calcium and ZAP70. The authors should discuss these data in the light of functional exhaustion, 12 hours of exposure to AML cells imprint the cells to reduced activation. Is this only by the CAR? Is TCR signaling normal?

Response:

We have modified this sentence to “We demonstrated that CAR T cells exposed to AML cells had impaired activation of calcium, ZAP70, ERK, and C-JUN”. In addition, we also discussed that although in exhausted T cells PD1 may provide an inhibitory signal to repress ZAP70 and ERK activity²¹, we did not observe increased expression of exhaustion markers such as PD1, TIM3, and LAG3 in AML-exposed CAR T cells during the limited observation

period. Whether prolonged exposure to AML can lead to exhaustion of CAR T cells needs to be further evaluated.

We have performed the stimulation with anti-CD3 and found that ERK, ZAP70, and C-JUN activation is also defective in AML pre-incubated CAR T cells, suggesting that TCR signaling is also partially impaired after AML pre-incubation.

We have added the related description in lines 149–152 page 8, in the Results section, and lines 328–329, 331–335, page 16, in the Discussion section of the revised manuscript.

17. Have the authors evaluate how overexpression of *cjun* affects tonic CAR signaling?

Response:

We have performed *in vitro* experiments to investigate the effect of C-JUN overexpression on tonic CAR signaling. Tonic signaling refers to CAR-induced antigen-independent T-cell activation²², which is primarily characterized by increased phosphorylation of CD3 ζ , increased expression of inhibitory receptors (PD-1, TIM-3, LAG-3), and an increased proportion of apoptotic cells, as observed in previous researches^{19,23}.

To determine the influence of C-JUN overexpression on tonic CAR signaling, we evaluated the expression levels of these markers. Our results showed that there was no significant difference in the proportion of apoptotic cells between the two groups. In addition, the expression of inhibitory receptors (PD-1, TIM-3, LAG-3) and the level of CD3 ζ phosphorylation were comparable between control and C-JUN CAR T cells. Based on these results, it can be speculated that C-JUN overexpression has no obvious effect on tonic CAR signaling.

We have added the data in the revised Figure 5I–K, and we have also added the description in lines 224–227, page 11, in the Results section of the revised manuscript.

Minor points:

1. the figures should be called in consecutive order in the manuscript

Response:

We have made the figures called in consecutive order in the revised manuscript.

2. The abbreviation c-JUN OE is not introduced

Response:

C-JUN OE means C-JUN-overexpressing CAR T cells, and now we have uniformly named C-JUN CAR T cells throughout the revised manuscript.

3. Lane 561; Phosphor-specific should be phosphor-specific

Response:

We have changed this to “phosphor-specific” in line 521, page 24, in the Methods section of the revised manuscript.

Reviewer #4 (Remarks to the Author); expert in clinical trial design and statistics:

The authors evaluate the safety of their proposed C-JUN-overexpressing CAR T cells in a Phase I trial, I have some comments on the clinical trial.

Response:

We are grateful for the reviewer’s comments. We hope that we addressed many of them in our revision and the response to the reviewer’s comments.

Major comments:

1. Based on the NCT04835519 registration, this study updated from a 3+3 design to a BOIN12 design, and the new design was planned to start on September 1, 2021, but in the paper, the authors stated that the four patients enrolled underwent CAR T cell infusion between April 13, 2021, and July 28, 2021. Were those 4 patients' clinical trials following the old protocol or the updated one? Please explain.

Response:

Thank you for raising this important point. Indeed, these four patients were enrolled following the old protocol using the “3+3” design. We have corrected the related description in line 645, page 29, in the Methods section of the revised manuscript.

2. How the decision to suspend this study was made, based on a predefined rule or other reason, the authors should provide the reason in the result.

Response:

Thank you for your question. We would like to clarify that the decision to stop the study was not based on any predetermined rules but was a decision made by IRB based on the clinical study protocol.

One of the main reasons was that the second patient developed a DLT event with grade 4 CRS, although the CRS in the latter two patients was well controlled with etanercept. In addition, one patient developed a serious infection. Despite stringent infection control measures and prompt medical intervention, the infection posed a significant risk to patient safety. Given the serious concerns about patient safety and the need to develop safer CAR T-cell products, we decided to suspend the clinical trial. We are planning to develop a new trial with an optimized strategy with these C-JUN-overexpressing CAR T cells.

We have added the related description in lines 300–302, page 14, in the Results section, and lines 420–425, pages 19–20, in the Discussion of the revised manuscript.

3. Since this study terminated too early, the objective defined in the protocol can not be evaluated based on the reported data, and the conclusion of "demonstrating efficacy in the first-in-human clinical trial" should be cautiously made.

Response:

We have changed the conclusion “demonstrated efficacy in first-in-human clinical trials” to “Optimized CD33 CAR T cells showed preliminary activity in the clinic, generating several

signs that support the further development of CAR T cell therapy for AML”.

We have added the related description in lines 433–434, page 20, in the Discussion section of the revised manuscript.

References:

1. Houtman, J.C., Houghtling, R.A., Barda-Saad, M., Toda, Y. & Samelson, L.E. Early phosphorylation kinetics of proteins involved in proximal TCR-mediated signaling pathways. *J Immunol* **175**, 2449-2458 (2005).
2. Perbellini, O., Cavallini, C., Chignola, R., Galasso, M. & Scupoli, M.T. Phospho-Specific Flow Cytometry Reveals Signaling Heterogeneity in T-Cell Acute Lymphoblastic Leukemia Cell Lines. *Cells* **11**(2022).
3. George, A.A., *et al.* Phosphoflow-Based Evaluation of Mek Inhibitors as Small-Molecule Therapeutics for B-Cell Precursor Acute Lymphoblastic Leukemia. *PLoS One* **10**, e0137917 (2015).
4. Love, M.I., Huber, W. & Anders, S. Moderated estimation of fold change and dispersion for RNA-seq data with DESeq2. *Genome Biol* **15**, 550 (2014).
5. Subramanian, A., *et al.* Gene set enrichment analysis: a knowledge-based approach for interpreting genome-wide expression profiles. *Proc Natl Acad Sci U S A* **102**, 15545-15550 (2005).
6. Stamm, H., *et al.* Immune checkpoints PVR and PVRL2 are prognostic markers in AML and their blockade represents a new therapeutic option. *Oncogene* **37**, 5269-5280 (2018).
7. Krenciute, G., *et al.* Transgenic Expression of IL15 Improves Antiglioma Activity of IL13R α 2-CAR T Cells but Results in Antigen Loss Variants. *Cancer Immunol Res* **5**, 571-581 (2017).
8. Mu-Mosley, H., *et al.* Transgenic Expression of IL15 Retains CD123-Redirected T Cells in a Less Differentiated State Resulting in Improved Anti-AML Activity in Autologous AML PDX Models. *Front Immunol* **13**, 880108 (2022).
9. Ataca Atilla, P., *et al.* Modulating TNF α activity allows transgenic IL15-Expressing CLL-1 CAR T cells to safely eliminate acute myeloid leukemia. *J Immunother Cancer* **8**(2020).
10. Tousley, A.M., *et al.* Co-opting signalling molecules enables logic-gated control of CAR T cells. *Nature* **615**, 507-516 (2023).
11. Sun, C., *et al.* THEMIS-SHP1 Recruitment by 4-1BB Tunes LCK-Mediated Priming

- of Chimeric Antigen Receptor-Redirected T Cells. *Cancer Cell* **37**, 216-225.e216 (2020).
12. Lynn, R.C., *et al.* c-Jun overexpression in CAR T cells induces exhaustion resistance. *Nature* **576**, 293-300 (2019).
 13. Batra, S.A., *et al.* Glypican-3-Specific CAR T Cells Coexpressing IL15 and IL21 Have Superior Expansion and Antitumor Activity against Hepatocellular Carcinoma. *Cancer Immunol Res* **8**, 309-320 (2020).
 14. Tambaro, F.P., *et al.* Autologous CD33-CAR-T cells for treatment of relapsed/refractory acute myelogenous leukemia. *Leukemia* **35**, 3282-3286 (2021).
 15. Wang, Q.S., *et al.* Treatment of CD33-directed chimeric antigen receptor-modified T cells in one patient with relapsed and refractory acute myeloid leukemia. *Mol Ther* **23**, 184-191 (2015).
 16. Lo, W.L., *et al.* Lck promotes Zap70-dependent LAT phosphorylation by bridging Zap70 to LAT. *Nat Immunol* **19**, 733-741 (2018).
 17. Schnoegl, D., Hiesinger, A., Huntington, N.D. & Gotthardt, D. AP-1 transcription factors in cytotoxic lymphocyte development and antitumor immunity. *Curr Opin Immunol* **85**, 102397 (2023).
 18. Gennert, D.G., *et al.* Dynamic chromatin regulatory landscape of human CAR T cell exhaustion. *Proc Natl Acad Sci U S A* **118**(2021).
 19. Long, A.H., *et al.* 4-1BB costimulation ameliorates T cell exhaustion induced by tonic signaling of chimeric antigen receptors. *Nat Med* **21**, 581-590 (2015).
 20. Li, S., *et al.* CD33-Specific Chimeric Antigen Receptor T Cells with Different Co-Stimulators Showed Potent Anti-Leukemia Efficacy and Different Phenotype. *Hum Gene Ther* **29**, 626-639 (2018).
 21. Fenwick, C., *et al.* T-cell exhaustion in HIV infection. *Immunol Rev* **292**, 149-163 (2019).
 22. Ajina, A. & Maher, J. Strategies to Address Chimeric Antigen Receptor Tonic Signaling. *Mol Cancer Ther* **17**, 1795-1815 (2018).
 23. Eyquem, J., *et al.* Targeting a CAR to the TRAC locus with CRISPR/Cas9 enhances tumour rejection. *Nature* **543**, 113-117 (2017).

REVIEWER COMMENTS

Reviewer #1 (Remarks to the Author):

Thank you for addressing my comments, this is a really nice piece of work.
I have one additional, but important comment/concern:

The abstract (line 51) states "... and three achieved complete remissions". From a clinical standpoint, this is simply not true, because complete remission in AML (using say the ELN 2017 criteria) requires normalization of blood counts. See table 6 in Dohner et al. Blood 2017;129:424-447. In fact, the most generous interpretation of the patients' clinical responses is "CR with incomplete hematologic recovery", which the authors do mention in the text (but not in the abstract). However, even that interpretation is, in my opinion, optimistic, because the marrow photomicrographs now provided in Suppl Fig 9 very clearly show that the marrow aspirates were grossly hemodilute. This means that the flow Cytometry results in Fig 8B are most likely representative of peripheral blood and not marrow. We aren't provided with marrow biopsy photomicrographs. It is crucial to make the distinction between (1) blast reduction (based on marrow aspirate differential count +/- flow on an appropriate specimen - i.e. not one that is hemodilute), (2) marrow aplasia (empty marrow that may manifest with a hemodilute aspirate smear and an empty marrow on biopsy, both with profound pancytopenia), or (3) CR +/- CRi as claimed by the authors.

This is important both for data and scientific integrity, as well as for overall interpretation of the CART cell trials in AML. I guess that 90% of people will read just the abstract (unfortunately), and it would be inappropriate for them to come away with the conclusion that this approach leads to 3/4 complete responses, or even CRi, since the data presented does not support that.

Since the paper is otherwise very good, and because I think it is important to add the clinical data to this preclinical paper, I would suggest that the authors reword the abstract and results sections to reflect my comments above perhaps by saying something like "... and blasts were no longer detectable in the marrow".

Reviewer #2 (Remarks to the Author):

The authors have responded to the reviewer's comments adequately. Additional experiments have been performed to clarify open issues and major changes in the manuscript have been implemented.

Reviewer #3 (Remarks to the Author):

I deeply thank the authors for answering seriously and very convincing to all my request.
Just a couple of very minor points:

Q1: are the statistical analysis done in figure 3J-L paired? The rationale behind my question is that the data seem to follow a pattern that can be donor dependent but a paired analysis could reflect statistical significances.

Q17, Figure 5J.

I am not sure whether it is a typo, but why were the cells stimulated with anti-CD3 antibodies in order to analyse tonic signal by the CAR? this has to be corrected or clarified.

Likewise, the flow cytometric analysis of phospho-z does not allow to distinguish between CAR-z or TCR-z. This compromises the interpretation of the results. Indeed, WB analysis would be more suitable to compare the tonic phosphorylation of the CAR-z vs TCR-z. Consider to delete the flow cytometric analysis or clearly state that they represent global z phosphorylation (both CAR and TCR) and therefore, they said nothing about CAR-induced tonic signalling.

Reviewer #4 (Remarks to the Author):

My concerns have been addressed, and I have no further comments on the clinical trial.

Point-by-point responses to feedback on Manuscript

Manuscript Number: NCOMMS-23-43280B

REVIEWER COMMENTS

Reviewer #1 (Remarks to the Author):

Thank you for addressing my comments, this is a really nice piece of work.

I have one additional, but important comment/concern:

The abstract (line 51) states "... and three achieved complete remissions". From a clinical standpoint, this is simply not true, because complete remission in AML (using say the ELN 2017 criteria) requires normalization of blood counts. See table 6 in Dohner et al. Blood 2017;129:424-447. In fact, the most generous interpretation of the patients' clinical responses is "CR with incomplete hematologic recovery", which the authors do mention in the text (but not in the abstract). However, even that interpretation is, in my opinion, optimistic, because the marrow photomicrographs now provided in Suppl Fig 9 very clearly show that the marrow aspirates were grossly hemodilute. This means that the flow Cytometry results in Fig 8B are most likely representative of peripheral blood and not marrow. We aren't provided with marrow biopsy photomicrographs. It is crucial to make the distinction between (1) blast reduction (based on marrow aspirate differential count +/- flow on an appropriate specimen - i.e. not one that is hemodilute), (2) marrow aplasia (empty marrow that may manifest with a hemodilute aspirate smear and an empty marrow on biopsy, both with profound pancytopenia), or (3) CR +/- CRi as claimed by the authors.

This is important both for data and scientific integrity, as well as for overall interpretation of the CART cell trials in AML. I guess that 90% of people will read just the abstract (unfortunately), and it would be inappropriate for them to come away with the conclusion that this approach leads to 3/4 complete responses, or even CRi, since the data presented does not support that.

Since the paper is otherwise very good, and because I think it is important to add the clinical data to this preclinical paper, I would suggest that the authors reword the abstract and results sections to reflect my comments above perhaps by saying something like "... and blasts were no longer detectable in the marrow".

Response:

We sincerely thank the reviewer for the thoughtful suggestions and kind reminder.

In this study, bone marrow aspiration and diagnosis procedures strictly follow standard clinical practice guidelines and were respectively conducted by authorized physicians and specialists in the diagnosis laboratory in our medical center. We have also discussed these issues with the senior specialists in our morphological and flow-cytometry diagnosis laboratory, and they have confirmed that the results in Suppl Fig 9A and Figure 8C were from bone marrow diagnosis. We have provided clearer, higher-resolution bone marrow micrographs in Suppl Fig 9A.

In addition, we observed the marrow aplasia in our patients, but the marrow aplasia resulted from severe myelosuppression following CD33 CAR-T therapy, rather than heavy contamination with peripheral blood cells. Indeed, in addition to the specific killing of CD33⁺ blasts, CAR T cells also eliminated normal myeloid CD33⁺ cells in the bone marrow. This may lead to myelosuppression and marrow aplasia. However, it should be noted that tumor cells may reappear as the bone marrow recovers. Following NCCN and ELN criteria, these patients were considered as “complete remissions with incomplete hematologic recovery (CRi)”.

We have modified the related description to “two patients had no detectable bone marrow blasts, and one had blast reduction after treatment” in lines 51–52, page 3, in the Abstract section, and we added the description “we also observed myelosuppression after CD33 CAR-T therapy” in lines 318–319, page 15, in the Results section of the revised manuscript.

Reviewer #2 (Remarks to the Author):

The authors have responded to the reviewer’s comments adequately. Additional experiments

have been performed to clarify open issues and major changes in the manuscript have been implemented.

Response:

We thank the reviewer for the review and comments on our manuscript.

Reviewer #3 (Remarks to the Author):

I deeply thank the authors for answering seriously and very convincing to all my request.

Just a couple of very minor points:

Q1: are the statistical analysis done in figure 3J-L paired? The rational behind my question is that the data seem to follow a pattern that can be donor dependent but a paired analysis could reflect statistical significances.

Response:

We thank the reviewer for pointing this out and helping us further improve our study.

In response to the review's suggestions, we have re-analyzed the data in Figure 3J-L using paired *t*-tests, and the results showed no significant differences in these panels. However, some variability between donors was observed, particularly with TIGIT receptor blockade ($p = 0.0631$).

Figure S4

After most of the revised manuscript was completed, and considering that the interaction of these receptors with CD155 is a central aspect of our study, we generated CAR T cells from an additional 1-2 new donors and repeated the experiments. The results from these additional donors also confirmed that there was no significant effect of blocking CD226, TIGIT, or CD96 on cytotoxicity. Due to time constraints and that we had nearly completed the editing of the first revision, we did not add this additional data to the figures.

Now, to make the conclusion more reliable, we presented the data from all the experiments we conducted. Regardless of whether we used paired or unpaired *t*-tests, the addition of new donors did not result in a statistically significant difference in the control and experimental groups.

Figure S4C

Figure 3J

We have updated the data in the revised Figure 3J and supplementary Figure 4C-E, and have added the related description in line 754, page 37, in the Figure legend section of the revised manuscript.

Q17, Figure 5J.

I am not sure whether it is a typo, but why were the cells stimulated with anti-CD3 antibodies in order to analyse tonic signal by the CAR? this has to be corrected or clarify.

Likewise, the flow cytometric analysis of phospho-z does not allow to distinguish between CAR-z or TCR-z. This compromise the interpretation of the results. Indeed, WB analysis would be more suitable to compare the tonic phosphorylation of the CAR-z vs TCR-z. Consider to delete the flow cytometric analysis or clearly stated that they represent global z phosphorylation (both CAR and TCR) and therefore, they said nothing about CAR-induced tonic signalling.

Response:

We greatly appreciate the reviewer's insightful comments.

We apologize for the incorrect use of anti-CD3 antibody stimulation. We also evaluated the tonic signal without anti-CD3 antibody stimulation and with PCDH T cells as a control (non-CAR transduction). There was an increase in CD3 ζ phosphorylation (p-CD3 ζ) in both control CAR T cells and C-JUN CAR T cells compared to PCDH T cells, but no difference was found between control CAR T cells and C-JUN CAR T cells. We agree with the reviewer that p-CD3 ζ levels represent global CD3 ζ phosphorylation, including both CAR and TCR. Therefore, we have modified the description to "It should be noted that both CAR and TCR can induce CD3 ζ phosphorylation, so these results suggest that C-JUN overexpression had little effect on the overall CD3 ζ signaling downstream of both TCR and CAR in the absence of overt antigen stimulation".

We have updated the data in the revised Figure 5K, and we have added the related description in lines 228–233, page 11, in the Results section, and in lines 796–798, page 41, in the Figure legend section of the revised manuscript.

Figure 5

K

Reviewer #4 (Remarks to the Author):

My concerns have been addressed, and I have no further comments on the clinical trial.

Response:

We thank the reviewer for the review and comments on our manuscript.

Figures

We have corrected some errors in the previous Supplementary Figure 7A (below, left) and changed it to the revised Supplementary Figure 7A (below, right) because we misplaced the figure, which is Supplementary Figure 6D. The description in the manuscript (Results section) is not erroneous and therefore does not require correction.